# A Few Large Shifts: Layer-Inconsistency Based Minimal Overhead Adversarial Example Detection

## Abstract

Deep neural networks (DNNs) are highly susceptible to adversarial examples–subtle, imperceptible perturbations that can lead to incorrect predictions. While detection-based defenses offer a practical alternative to adversarial training, many existing methods depend on external models, complex architectures, or adversarial data, limiting their efficiency and generalizability. We introduce a lightweight, plug-in detection framework that leverages internal layer-wise inconsistencies within the target model itself, requiring only benign data for calibration. Our approach is grounded in the **A Few Large Shifts Assumption**, an empirical hypothesis that adversarial perturbations often induce large, localized violations of *layer-wise Lipschitz continuity* in a small subset of adjacent layer transitions. Building on this, we propose two complementary strategies–**Recovery Testing (RT)** and **Logit-layer Testing (LT)**–to measure intermediate-layer and logit-layer inconsistencies, and fuse them through RLT. Evaluated on CIFAR-10, CIFAR-100, and ImageNet under standard and adaptive threat models, our method achieves strong detection performance with substantially lower overhead than detector families requiring external encoders or reference-set retrieval. Furthermore, our system-level analysis provides a practical threshold-selection rule with a stated lower bound on system accuracy under the metric assumptions used in the analysis.

## 1 Introduction

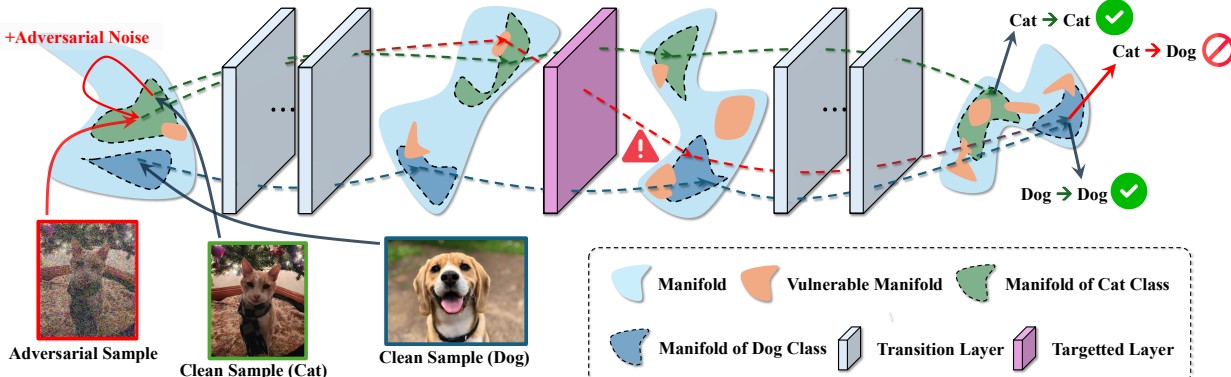

Figure 1: **Illustration of A Few Large Shifts Assumption**–an assumption on a few large perturbation shifts on a simple target classifier causes vulnerable manifolds that can trigger an unexpected transition to a different manifold space. Leveraging these vulnerable regions, an adversarial sample is crafted using an attack algorithm that applies perturbation noise, causing the original sample to shift from the cat class manifold to the dog class manifold through transitions occurring at specific targeted layers.

Deep neural networks (DNNs) have been broadly deployed in computer vision, natural language processing, multi-modal tasks, and beyond Nan et al. (2024); Khachatryan et al. (2023); Zhang et al. (2024). Although they exhibit remarkable performance, *adversarial examples (AEs)*—inputs containing small yet malicious

perturbations—can induce misclassifications while appearing virtually unchanged to human observers Demontis et al. (2019). This vulnerability poses severe risks in high-stakes domains such as autonomous driving or disease diagnosis Zheng et al. (2024); Ravikumar et al. (2024), underlining the need for robust defenses.

Existing defensive strategies generally fall into three categories: adversarial training, input purification, and AE detection He et al. (2022); Abusnaina et al. (2021). *Adversarial training*, re-trains or fine-tunes a model using adversarially perturbed samples to have Adversarially Trained Classifier (ATC) Wang et al. (2024); Liu et al. (2024), despite its robustness Elfwing et al. (2018); Zhang et al. (2019), it incurs high computational costs and may compromise clean-data accuracy. *Input purification* methods Mao et al. (2021); Song et al. (2024) attempt to remove adversarial noise through preprocessing (e.g., denoising), yet often fail against adaptive attacks Croce & Hein (2020).

A more flexible alternative is AE detection, which avoids the costly retraining process of ATCs by instead rejecting suspicious inputs at test time Xu et al. (2017); Zuo & Zeng (2021). Detection-based defenses offer distinct advantages, including *lower implementation costs*, as they do not require exhaustive adversarial training with adversarial data, and *tunable robustness*, where sensitivity can be adjusted to meet application-specific accuracy–robustness trade-offs. Furthermore, their *plug-and-play* nature allows them to be integrated with existing models, including ATCs, to further enhance system-level robustness. By architecturally separating the detection mechanism from the classifier, this approach introduces an additional layer of security at the system level. AE detection techniques are often categorized into two types: those that compare inputs to a reference set (e.g., Deep k-Nearest Neighbors, DkNN Papernot & McDaniel (2018); Latent Neighborhood Graph, LNGAbusnaina et al. (2021)), and those that analyze invariants in the learned representations Jiang et al. (2020); Chen & He (2021). Despite these benefits, prominent detection paradigms suffer from practical limitations. Reference-based detectors often require storing adversarial examples or constructing complex neighbor graphs, which is computationally intensive. To address this, recent work Zhiyuan et al. (2024) has proposed detecting consistency between augmented inputs using pre-trained Self-Supervised Learning (SSL) models. However, this introduces significant overhead from large external models—which can be over *22 times larger* than the target classifier—and assumes the availability of high-quality, domain-specific SSL models.

To eliminate the overhead of external models, complex data structures, or heavy augmentations, we develop a self-contained, layer-wise detector that scrutinizes the network's own representations. This raises a central question: *How do adversarial perturbations propagate through a deep neural network?* Prior work suggests that such perturbations often leave early-layer features largely intact while inducing sharp deviations in deeper layers. Motivated by these findings, our empirical analysis investigates this behavior directly (Subsection 3.4 and Appendix L). As visualized in Figure 3 and further quantified in Figure 6–Table 32, we observe that while benign perturbations produce relatively flat error profiles across layers, whereas adversarial perturbations induce large, localized shifts at only a few critical points in the network. Rather than contrasting these deviations with external references, we ask a different core question: *Do these internal feature jumps themselves expose adversarial inputs?* We formalize this intuition in the **A Few Large Shifts Assumption**, which posits that adversarial perturbations produce large, localized shifts between a few critical layers. To quantify the observed localized shifts, we introduce the concept of *layer-wise Lipschitz continuity*, defining these shifts as localized violations of layer-wise Lipschitz continuity. Guided by this principle, our framework is the first designed to empirically measure these violations using a detector calibrated only on benign data. As illustrated in Figure 1, these sparse disruptions leave a detectable footprint across successive layers, which we capture using a lightweight detection framework grounded in layer-wise inconsistency with two complementary probes: Recovery Testing (RT) and Logit-layer Testing (LT). We then fuse their signals through Recovery-and-Logit Testing (RLT), aligning their scores via quantile normalization to robustly flag a broad spectrum of attacks, including those that evade one probe in isolation.

The distinguishing features and requirements of our method are summarized in Table 1, where we compare RT and LT against several representative baselines. As shown, our method is self-sufficient (requires no adversarial examples), model-local (requires no external SSL), and low-overhead (no kNN graphs or excessive augmentation), while still requiring benign-only auxiliary calibration for RT/LT. Furthermore, while prior AE detection works often omit system-level operating-point analysis, our work addresses this gap by introducing

a threshold-selection method with an explicit lower bound on overall system accuracy under the stated metric assumptions, as detailed in Appendix H.

Our key contributions are as follows:

- We propose a novel adversarial detection paradigm that exploits partial consistency in internal feature transformations of adversarial examples. Unlike prior methods that rely on computationally expensive external reference sets or large auxiliary models, our approach eliminates this overhead by measuring inconsistencies across a model's own layers for a single input.
- We are the first to formalize these internal disruptions as violations of *layer-wise Lipschitz continuity*. We introduce two complementary probes, **Recovery Testing (RT)** and **Logit-layer Testing (LT)**, as practical and efficient methods to empirically quantify these violations using only benign data.
- We conduct a comprehensive evaluation on standard benchmarks, demonstrating competitive detection performance against a wide range of threats, including adaptive attacks and generalization to various architectures. Crucially, our system-level analysis yields a practical method for selecting a detection threshold with a **guaranteed lower bound** on system accuracy under the analysis assumptions, enhancing the method's practical reliability.

Table 1: Comparison of Requirements / Properties Across Methods. **RT** = Recovery Testing, **LT** = Logit-layer Testing. We also include **BEYOND** Zhiyuan et al. (2024), **LID** Ma et al. (2018), **Mao** Mao et al. (2021), **Hu** Hu et al. (2019), **DkNN** Papernot & McDaniel (2018), **kNN-Def.** Dubey et al. (2019). A ✓ indicates that the method satisfies the criterion, while a ✗ indicates it does not.

| | Ours | | Baseline Methods | | | | | |
|---|---|---|---|---|---|---|---|---|
| Criterion | RT | LT | BEYOND | LID | Mao | Hu | DkNN | kNN-Def. |
| Self-Sufficient Data? (No Adversarial Data Needed?) | ✓ | ✓ | ✓ | ✗ | ✗ | ✗ | ✗ | ✗ |
| Standalone Model? (No Extra Pre-trained Model Dependencies?) | ✓ | ✓ | ✗ | ✓ | ✗ | ✓ | ✓ | ✓ |
| No Heavy kNN Retrieval? (No Nearest-Neighbor Search Overhead?) | ✓ | ✓ | ✓ | ✗ | ✓ | ✓ | ✗ | ✗ |
| No Excessive Augmentations? (No Many Augmentations?) | ✓ | ✓ | ✗ | ✓ | ✓ | ✓ | ✓ | ✓ |
| No Extra Optimization? (No Additional Training Needed?) | ✗ | ✓/✗ | ✗ | ✗ | ✗ | ✗ | ✗ | ✗ |

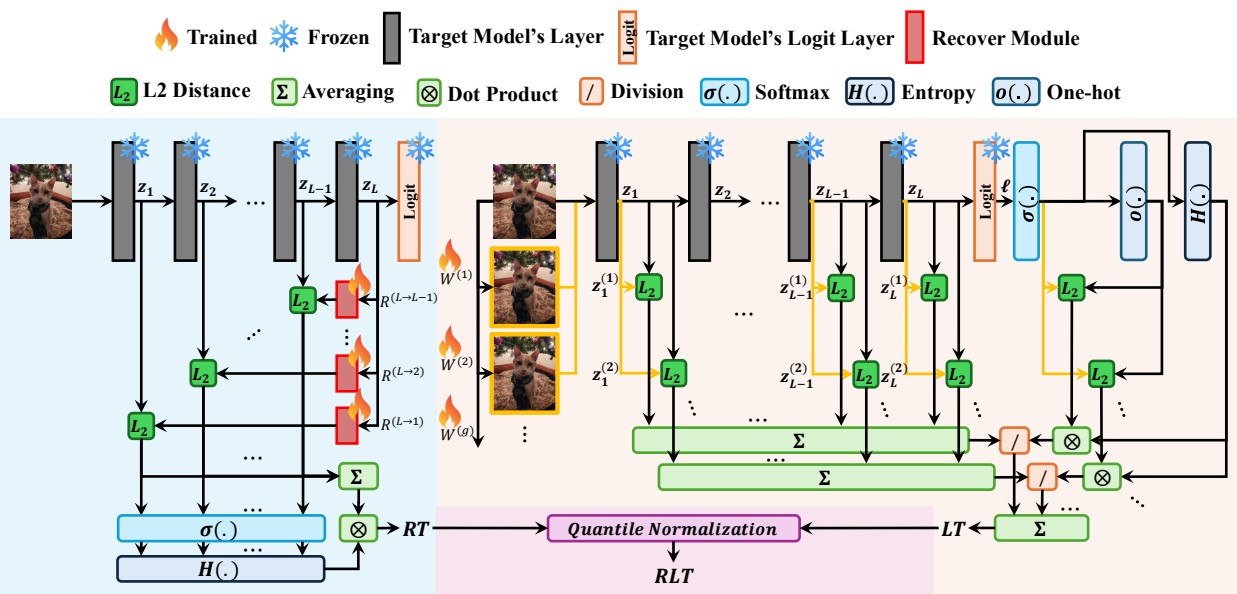

Figure 2: **Overview of our layer-wise adversarial detection framework.** (Left) *Recovery Testing (RT)* trains regressors $R^{(L \to k)}$ to reconstruct intermediate features $z_k$ from the final embedding $z_L$; detection is based on the entropy-weighted reconstruction error $\|z_k - R^{(L \to k)}(z_L)\|_2^2$. (Right) *Logit-layer Testing (LT)* applies perturbations $W^{(g)}$ to compute feature and logit discrepancies, $\Delta z^{(g)} = \|z_i - z_i^{(g)}\|_2^2$ and $\Delta \ell^{(g)} = \|o(\sigma(\ell)) - \sigma(\ell^{(g)})\|_2^2$, which are combined as $\frac{H(\sigma(\ell))\Delta \ell^{(g)}}{\Delta z^{(g)}}$. Final RT and LT scores are averaged over $g$. (Bottom) *Recovery and Logit Testing (RLT)* integrates RT and LT scores using a linear combination after quantile normalization.

## 2 Methodology

In this section, we introduce our layer-wise adversarial example detection framework (Figure 2), comprising three measures: RT, LT, and their fusion RLT. RT captures inconsistencies in intermediate-layer embeddings, while LT measures logit instability under augmented inputs. At test time, pick a detection measure $m \in \{\text{RT}, \text{LT}, \text{RLT}\}$ and threshold $\tau_m$. We declare $x$ *adversarial* iff $m(x) > \tau_m$, i.e., $\hat{a}(x) = \mathbb{I}\{m(x) > \tau_m\} \in \{0, 1\}$. Details of the selection of each threshold $\tau_m$ are delineated in Appendix H. The algorithm for calculating each measure is shown in Algorithm 1. Theoretical assumptions and proofs of our method are shown in Appendix B and Appendix C, respectively.

### 2.1 Definition and Notations

Given a target DNN $f(\cdot) = f_{logit} \circ f_L \circ f_{L-1} \circ \cdots \circ f_1(\cdot)$ with $L$ intermediate layers $f_i$ and one logit layer $f_{logit}$ at the end, we compute intermediate representations $z_i(x_j) = f_i \circ \cdots \circ f_1(x_j) \in \mathbb{R}^{D_i}$ for all $i \in \{1, 2, \cdots, L\}$, where $D_i$ denotes the dimensionality of the embedding space and $x_j$ denotes an input sample. We assume the logit layer takes the last intermediate embedding $z_L$ to produce the final logit output $\ell = f_{logit}(z_L) \in \mathbb{R}^C$, where $C$ is the number of output classes. We denote the softmax function as $\sigma(x) = \frac{\exp x}{\mathbf{1}^\top \exp x}$, the Shannon entropy as $\mathcal{H}(p) = -\sum_i p_i \log(p_i + \varepsilon)$ with a small numerical constant $\varepsilon > 0$, and the class-$c$ one-hot basis vector as $\mathbf{o}_c \in \{0, 1\}^C$. Labels are not used to train the auxiliary modules: all recovery modules, transformations, and thresholds are fitted/calibrated using only benign samples $\mathcal{D}_{norm} = \{x_1, x_2, \cdots, x_N\}$. At test time, LT uses the model-predicted class $\hat{y} = \arg\max_c \sigma_c(\ell(x))$, not the ground-truth label.

### 2.2 Core Assumption

We begin by formalizing the central structural assumption underlying our detection framework:

**Assumption 1 (A Few Large Shifts).** *Let $x^{\text{adv}} = x + \delta$ be an adversarial input, and let $z_k(x)$ denote the output of $k$-th layer $f_k$. We assume there exists a small subset of layers $\mathcal{T} \subseteq \{f_1, \ldots, f_L, f_{logit}\}$ such that for any $f_k \in \mathcal{T}$,*

$$\frac{1}{D_{k+1}}\|z_{k+1}(x^{\text{adv}}) - z_{k+1}(x)\|_2^2 \gg \frac{1}{D_k}\|z_k(x^{\text{adv}}) - z_k(x)\|_2^2.$$

*For all other layers $f_j \notin \mathcal{T}$, the shift between representations remains small:*

$$\frac{1}{D_{j+1}}\|z_{j+1}(x^{\text{adv}}) - z_{j+1}(x)\|_2^2 \approx \frac{1}{D_j}\|z_j(x^{\text{adv}}) - z_j(x)\|_2^2.$$

This assumption formalizes the intuition that adversarial perturbations cause disproportionately large changes in representation between specific adjacent layers while most other transitions remain relatively stable—a phenomenon that can be quantified based on Lipschitz continuity, which we term **layer-wise Lipschitz continuity** (See Appendix B.1). Specifically, the condition $\|\mathbf{z}_{k+1}(x') - \mathbf{z}_{k+1}(x)\|_2 \gg \|\mathbf{z}_k(x') - \mathbf{z}_k(x)\|_2$ implies a large local Lipschitz constant for the transformation between layers, indicating a region of high instability. The following sections introduce our proposed testing measures, which are designed to empirically identify and quantify these localized violations of layer-wise Lipschitz continuity.

### 2.3 Recovery Testing Measure

For each hidden layer $k \in \{k_{RT}, k_{RT} + 1, \ldots, L - 1\}$, we train an approximate inverse function: $R^{(L \to k)} : \mathbb{R}^{D_L} \longrightarrow \mathbb{R}^{D_k}$ to reconstruct $z_k$ from $z_L$ with hyperparameter $k_{RT}$. Each $R^{(L \to k)}$ is implemented as a lightweight MLP with 3–4 layers and trained by a mean squared error loss:

$$\mathcal{L}_{RT} = \frac{1}{N}\sum_{n=1}^{N}\sum_{k=k_{RT}}^{L-1}\| z_k(x_n) - R^{(L \to k)}(z_L(x_n))\|_2^2. \tag{1}$$

At test time, we define the normalized squared error only for the layers for which a recovery module is trained, $k \in \{k_{RT}, \ldots, L - 1\}$, as

$$e_k(x) = \| z_k(x) - R^{(L \to k)}(z_L(x))\|_2^2, \quad k = k_{RT}, \ldots, L - 1. \tag{2}$$

Collecting the vector $e_{RT}(x) = (e_{k_{RT}}(x), \ldots, e_{L-1}(x))$, we normalize it with softmax to obtain a distribution $\sigma(e_{RT}(x))$ across the recovered layers. The non-uniformity of this distribution is captured using

inverse entropy $\log(L - k_{RT}) - \mathcal{H}(\sigma(e_{RT}(x)))$. The final score is computed by taking the average of the raw reconstruction errors and weighting it by this information-based term:

$$RT(x) = \left(\log(L - k_{RT}) - \mathcal{H}(\sigma(e_{RT}(x)))\right) \log\left(\frac{1}{L - k_{RT}} \sum_{k=k_{RT}}^{L-1} e_k(x)\right). \tag{3}$$

We apply a logarithm to the average of raw errors for numerical stability. Under our assumption, adversarial samples that target specific layers $\mathcal{T} \subseteq \{f_1, \ldots, f_{L-1}\}$ will yield higher $RT(x)$ values due to large reconstruction errors (see Theorem 1) and a sharply peaked error distribution. Intuitively, the first term of Equation 3 measures how deterministic the error distribution is across layers, while the second term captures the magnitude of **a few large shifts**.

## 2.4 Logit-layer Testing Measure

Although RT effectively detects perturbations in intermediate layers $f_i$, it cannot be applied to the final logit layer $f_{\text{logit}}$, since there is no subsequent representation from which to reconstruct the logits. To address this—and motivated by the intuition that adversarial examples aim to induce misclassification—we introduce Logit-layer Testing (LT): we quantify uncertainty at $f_{\text{logit}}$, relative to changes in intermediate features, using data-driven, low-cost input augmentations.

Let $\{W^{(g)}\}_{g=1}^G$ be a small set (e.g., $1 \leq G \leq 6$) of image transformation matrices. Each matrix $W^{(g)}$ is initialized as an identity transformation and is then fine-tuned on benign data using gradient descent. The LT score itself, as defined in Equation 2.4, serves as the loss function for this process, encouraging the transformations to find augmentations that preserve logit stability for benign inputs, i.e., $\sigma(f(x)) \approx \sigma(f(W^{(g)}x))$, while still perturbing intermediate features. This establishes a consistent baseline against which adversarial instability can be measured.

Given a test input, for each augmentation $g$, we measure two types of inconsistencies between the original and augmented inputs: feature-space drift $\Delta z^{(g)}(x|W^{(g)})$ and change in logit decidedness $\Delta\ell^{(g)}(x|W^{(g)})$. The feature-space drift is measured by averaging the $L_2$ distances across all intermediate outputs: $\Delta z^{(g)}(x|W^{(g)}) = \frac{1}{L - k_{LT} + 1} \sum_{i=k_{LT}}^{L} \|z_i(x) - z_i(W^{(g)}x)\|_2^2$ with hyperparameter $k_{LT}$. The logit decidedness is measured by computing the $L_2$ distance between the one-hot vector of the original predicted class and the augmented-input softmax output: $\Delta\ell^{(g)}(x|W^{(g)}) = \|\mathbf{o}_{\hat{y}} - \sigma(\ell(W^{(g)}x))\|_2^2$, where $\hat{y} = \arg\max_c \sigma_c(\ell(x))$. We then combine the two quantities into $\frac{\Delta\ell^{(g)}(x|W^{(g)})}{\Delta z^{(g)}(x|W^{(g)})}$, weight it by the entropy of the logit score vector $\mathcal{H}(\sigma(\ell(x)))$ to emphasize inputs closer to the decision boundary Galil & El-Yaniv (2021), and finally average over all augmentations as follows:

$$LT(x) = \frac{1}{G} \sum_{g=1}^G \log\left(\mathcal{H}(\sigma(\ell(x)))\Delta\ell^{(g)}(x|W^{(g)}) + \varepsilon\right) - \log\left(\Delta z^{(g)}(x|W^{(g)}) + \varepsilon\right). \tag{4}$$

As before, the logarithm ensures numerical stability. Intuitively, under a successful adversarial attack the logit shift $\Delta\ell^{(g)}$ becomes disproportionately large compared to the accumulated feature drift $\Delta z^{(g)}$, which is precisely what LT is designed to detect (Theorem 3).

To fine-tune the transformation matrices $\{W^{(g)}\}$, we directly use $LT(x)$ as the training loss:

$$\mathcal{L}_{LT} = \frac{1}{N} \sum_{n=1}^N LT(x_n). \tag{5}$$

## 2.5 Recovery and Logit Testing Combined Measure

To capture inconsistencies in both intermediate and logit layers, we introduce a combined score called Recovery and Logit Testing. This score integrates RT and LT while correcting for differences in their statistical distributions using quantile normalization. Specifically, we transform each score based on its empirical cumulative distribution estimated from the benign training set, and map it to the standard normal distribution.

Let $\hat{\mathcal{F}}_{RT}$ and $\hat{\mathcal{F}}_{LT}$ be the empirical cumulative distribution functions (CDFs) of RT and LT scores computed over the benign data. Let $\Phi^{-1}$ denote the quantile function (inverse CDF) of the standard normal distribution. At test time, each score is transformed as follows:

Table 2: The AUC of Different Adversarial Detection Approaches on CIFAR-10. The results are the mean and standard deviation of 5 runs. Our methods are included for comparison. Classifier: ResNet110, FGSM: $\epsilon = 0.05$, PGD: $\epsilon = 0.02$. Note that our methods and BEYOND need no AE for training, leading to the same value on both seen and unseen settings. The bolded values are the best performance, and the underlined italicized values are the second-best performance.

| Method | Unseen (Attacks in training are excluded from tests) | | | | Seen (Attacks in training are included in tests) | | | | |
|---|---|---|---|---|---|---|---|---|---|
| | **FGSM** | **PGD** | **AutoAttack** | **Square** | **FGSM** | **PGD** | **CW** | **AutoAttack** | **Square** |
| DkNN Papernot & McDaniel (2018) | 61.55 ±0.023 | 51.22 ±0.026 | 52.12 ±0.023 | 59.46 ±0.022 | 61.55 ±0.023 | 51.22 ±0.026 | 61.52 ±0.028 | 52.12 ±0.023 | 59.46 ±0.022 |
| kNN Dubey et al. (2019) | 61.83 ±0.018 | 54.52 ±0.022 | 52.67 ±0.022 | 73.39 ±0.020 | 61.83 ±0.018 | 54.52 ±0.022 | 62.23 ±0.019 | 52.67 ±0.022 | 73.39 ±0.020 |
| LID Ma et al. (2018) | 71.08 ±0.024 | 61.33 ±0.025 | 55.56 ±0.021 | 66.18 ±0.025 | 73.61 ±0.020 | 67.98 ±0.020 | 55.68 ±0.021 | 56.33 ±0.024 | 85.94 ±0.018 |
| Hu Hu et al. (2019) | 84.51 ±0.025 | 58.59 ±0.028 | 53.55 ±0.029 | 95.82 ±0.020 | 84.51 ±0.025 | 58.59 ±0.028 | 91.02 ±0.022 | 53.55 ±0.029 | 95.82 ±0.020 |
| Mao Mao et al. (2021) | 95.33 ±0.012 | 82.61 ±0.016 | 81.95 ±0.020 | 85.76 ±0.019 | 95.33 ±0.012 | 82.61 ±0.016 | 83.10 ±0.018 | 81.95 ±0.020 | 85.76 ±0.019 |
| LNG Abusnaina et al. (2021) | 98.51 | 63.14 | 58.47 | 94.71 | _99.88_ | 91.39 | 89.74 | 84.03 | _98.82_ |
| BEYOND Zhiyuan et al. (2024) | 98.89 ±0.013 | _99.28_ ±0.020 | 99.16 ±0.021 | 99.27 ±0.016 | 98.89 ±0.013 | _99.28_ ±0.020 | 99.20 ±0.008 | 99.16 ±0.021 | **99.27** ±0.016 |
| **Our Approaches** | | | | | | | | | |
| RT | **99.93** ±0.005 | 96.89 ±0.071 | **99.99** ±0.000 | 85.38 ±0.344 | **99.93** ±0.005 | 96.89 ±0.071 | **99.99** ±0.002 | **99.99** ±0.000 | 85.38 ±0.344 |
| LT | 97.50 ±0.038 | 98.61 ±0.042 | _99.60_ ±0.018 | _97.47_ ±0.036 | 97.50 ±0.038 | 98.61 ±0.042 | 97.08 ±0.027 | _99.60_ ±0.018 | 97.47 ±0.036 |
| RLT | _99.85_ ±0.005 | **99.37** ±0.011 | **99.99** ±0.000 | 95.95 ±0.102 | 99.85 ±0.005 | **99.37** ±0.011 | _99.91_ ±0.004 | **99.99** ±0.000 | 95.95 ±0.102 |

Table 3: The AUC of Different Adversarial Detection Approaches on ImageNet. To align with baselines, classifier: DenseNet121, FGSM: $\epsilon = 0.05$, PGD: $\epsilon = 0.02$. Due to memory and resource constraints, baseline methods are not evaluated against AutoAttack on ImageNet.

| Method | Unseen | | Seen | | |
|---|---|---|---|---|---|
| | **FGSM** | **PGD** | **FGSM** | **PGD** | **CW** |
| DkNN Papernot & McDaniel (2018) | 89.16 ±0.038 | 78.00 ±0.041 | 89.16 ±0.038 | 78.00 ±0.041 | 68.91 ±0.044 |
| kNN Dubey et al. (2019) | 51.63 ±0.04 | 51.14 ±0.039 | 51.63 ±0.04 | 51.14 ±0.039 | 50.73 ±0.04 |
| LID Ma et al. (2018) | 90.32 ±0.046 | 52.56 ±0.038 | _99.24_ ±0.043 | 98.09 ±0.042 | 58.83 ±0.041 |
| Hu Hu et al. (2019) | 72.56 ±0.037 | 86.00 ±0.042 | 72.56 ±0.037 | 86.00 ±0.042 | 80.79 ±0.044 |
| LNG Abusnaina et al. (2021) | 96.85 | 89.61 | **99.53** | _98.42_ | 86.05 |
| BEYOND Zhiyuan et al. (2024) | _97.59_ ±0.04 | 96.26 ±0.045 | 97.59 ±0.04 | 96.26 ±0.045 | **95.46** ±0.047 |
| **Our Approaches** | | | | | |
| RT | 94.31 ±0.457 | **99.99** ±0.000 | 94.31 ±0.457 | **99.99** ±0.000 | 92.18 ±0.135 |
| LT | 96.18 ±0.028 | _97.89_ ±0.021 | 96.18 ±0.028 | 97.89 ±0.021 | _94.06_ ±0.215 |
| RLT | **97.60** ±0.048 | **99.99** ±0.000 | 97.60 ±0.048 | **99.99** ±0.000 | 91.19 ±0.022 |

$$RT_{norm}(x) = \Phi^{-1}\left(\hat{\mathcal{F}}_{RT}(RT(x))\right), \quad LT_{norm}(x) = \Phi^{-1}\left(\hat{\mathcal{F}}_{LT}(LT(x))\right).$$

The final RLT score is computed by summing the squared normalized values:

$$RLT(x) = (RT_{norm}(x))^2 + (LT_{norm}(x))^2. \tag{6}$$

This quantile-based transformation aligns both RT and LT scores to a common standard normal distribution, ensuring that the final combined score reflects significant deviations under either test (Theorem 4), independent of their original scales. This enhances robustness to varied attack types and scoring dynamics.

## 3  Experiments

Table 4: RA(%) under Orthogonal-PGD Adaptive Attack using CIFAR-10 and ResNet110.

| Defense | $L_\infty = 0.01$ | | $L_\infty = 8/255$ | |
|---|---|---|---|---|
| | **RA@FPR5%** | **RA@FPR50%** | **RA@FPR5%** | **RA@FPR50%** |
| **Ours (RLT)** | _75.40_ | **99.58** | **33.70** | **80.77** |
| BEYOND Zhiyuan et al. (2024) | **88.38** | _98.81_ | _13.80_ | _48.20_ |
| Trapdoor Shan et al. (2020) | 0.00 | 7.00 | 0.00 | 8.00 |
| DLA Sperl et al. (2020) | 62.60 | 83.70 | 0.00 | 28.20 |
| SID Tian et al. (2021) | 6.90 | 23.40 | 0.00 | 1.60 |
| SPAM Liu et al. (2019) | 1.20 | 46.00 | 0.00 | 38.00 |

### 3.1  Experimental Setup

**Datasets.** We evaluate our method on CIFAR-10, CIFAR-100, and ImageNet. CIFAR-10 and CIFAR-100 each contain 60,000 $32 \times 32$ images with standard training (50k) and test (10k) splits, across 10 and 100 classes respectively. For ImageNet, we use the official training and validation sets, resizing all images to $256 \times 256$ and applying standard normalization. CIFAR-10 is used for the main comparison to prior detectors, CIFAR-100 evaluates fine-grained class scaling, and ImageNet serves as the large-scale benchmark.

**Backbone Models.** For the main detector comparison, we use the same backbones as the baseline protocols: ResNet-110 for CIFAR-10 and DenseNet-121 for ImageNet. For the additional assumption validation in

Table 5: RA (%) scores of end-to-end attack with PGD under perturbation $\epsilon = 8/255$ (ResNet110, CIFAR10). The results are the mean and standard deviation of 5 runs.

| $\lambda$ | @FPR5% | @FPR10% | @FPR15% | @FPR20% | @FPR25% | @FPR30% | @FPR35% | @FPR40% | @FPR45% | @FPR50% |
|---|---|---|---|---|---|---|---|---|---|---|
| 1.00 | $20.72_{\pm 8.04}$ | $25.96_{\pm 6.88}$ | $32.05_{\pm 5.72}$ | $38.23_{\pm 5.11}$ | $44.48_{\pm 4.52}$ | $50.15_{\pm 4.13}$ | $55.15_{\pm 3.56}$ | $60.09_{\pm 3.10}$ | $64.54_{\pm 2.42}$ | $68.53_{\pm 2.21}$ |
| 0.50 | $22.64_{\pm 4.46}$ | $30.45_{\pm 3.25}$ | $37.96_{\pm 2.08}$ | $44.78_{\pm 1.81}$ | $51.19_{\pm 1.30}$ | $56.41_{\pm 0.98}$ | $61.21_{\pm 0.96}$ | $65.44_{\pm 0.96}$ | $69.35_{\pm 0.83}$ | $73.00_{\pm 0.92}$ |
| 0.25 | $23.39_{\pm 1.76}$ | $32.92_{\pm 0.49}$ | $40.88_{\pm 1.04}$ | $48.04_{\pm 1.13}$ | $54.32_{\pm 1.58}$ | $59.37_{\pm 1.44}$ | $63.90_{\pm 1.82}$ | $67.98_{\pm 1.82}$ | $71.51_{\pm 1.89}$ | $74.83_{\pm 1.85}$ |
| Mao Mao et al. (2021) | | | | | 18.97 | | | | | |
| BEYOND Zhiyuan et al. (2024) | | | | | 19.45 | | | | | |

Table 6: RA (%) scores of end-to-end attack with PGD under perturbation $\epsilon = 8/255$ (ResNet110, CIFAR10) when using an ATC. The results are the mean and standard deviation of 5 runs.

| $\lambda$ | @FPR5% | @FPR10% | @FPR15% | @FPR20% | @FPR25% | @FPR30% | @FPR35% | @FPR40% | @FPR45% | @FPR50% |
|---|---|---|---|---|---|---|---|---|---|---|
| 1.00 | $93.32_{\pm 0.13}$ | $93.38_{\pm 0.14}$ | $93.41_{\pm 0.11}$ | $93.43_{\pm 0.09}$ | $93.97_{\pm 0.12}$ | $94.49_{\pm 0.15}$ | $94.99_{\pm 0.14}$ | $95.50_{\pm 0.13}$ | $95.93_{\pm 0.14}$ | $96.32_{\pm 0.11}$ |
| 0.50 | $93.53_{\pm 0.15}$ | $94.17_{\pm 0.12}$ | $94.27_{\pm 0.11}$ | $94.33_{\pm 0.09}$ | $94.86_{\pm 0.13}$ | $95.41_{\pm 0.09}$ | $95.88_{\pm 0.09}$ | $96.27_{\pm 0.11}$ | $96.65_{\pm 0.14}$ | $97.00_{\pm 0.15}$ |
| 0.25 | $93.57_{\pm 0.14}$ | $94.46_{\pm 0.16}$ | $94.60_{\pm 0.13}$ | $94.85_{\pm 0.07}$ | $95.40_{\pm 0.05}$ | $95.87_{\pm 0.08}$ | $96.32_{\pm 0.14}$ | $96.71_{\pm 0.12}$ | $97.04_{\pm 0.12}$ | $97.30_{\pm 0.14}$ |
| Mao Mao et al. (2021) | | | | | 75.09 | | | | | |
| BEYOND Zhiyuan et al. (2024) | | | | | 93.20 | | | | | |

Appendix L, we evaluate downloaded torchvision-style pretrained backbones where available across ResNet-18, ResNet-34, VGG-11, and VGG-13 on CIFAR-10, CIFAR-100, and ImageNet. In all cases, we extract intermediate features without fine-tuning the backbone, ensuring a consistent foundation for detection.

**Threat Models.** We evaluate our detection framework under two standard adversarial settings: *Limited Knowledge* and *Perfect Knowledge*, following the protocol of previous works Apruzzese et al. (2023); Zhiyuan et al. (2024). In the *Limited Knowledge* setting, the adversary has full access to the target classifier but is unaware of the detection mechanism, which remains confidential. In contrast, the *Perfect Knowledge* (adaptive attack) setting assumes that the adversary has full knowledge of both the classifier and the detection strategy, enabling it to craft attacks specifically to evade detection.

**Attack Methods.** We evaluate robustness and detection performance under a diverse suite of standard attacks, including Fast Gradient Sign Method (FGSM), Projected Gradient Descent (PGD), Carlini–Wagner (CW), AutoAttack, and Square Attack Andriushchenko et al. (2020). To assess adaptive evasion, we also include Orthogonal-PGD Bryniarski et al. (2021) and end-to-end PGD objectives that explicitly combine misclassification with detector-score suppression. These attacks provide a strong adaptive evaluation, while we avoid claiming coverage under all possible adaptive strategies.

**Implementation Details.** All models are implemented in PyTorch and trained on NVIDIA RTX 6000 Ada. We use a batch size of 32, AdamW optimizer with learning rate $1 \times 10^{-4}$ and weight decay of 0.01, training the recover modules and augmentation matrices for 50 epochs.

## 3.2 Detection Performance under Standard Attacks

We evaluate the effectiveness of our proposed detection scores, namely RT, LT, and RLT, on CIFAR-10 and ImageNet under standard adversarial threat models, including FGSM, PGD, CW, AutoAttack, and Square Attack. We benchmark against established baselines such as LID Ma et al. (2018), DkNN Papernot & McDaniel (2018), LNG Abusnaina et al. (2021), and the recent SSL-based BEYOND Zhiyuan et al. (2024). The Area Under the Receiver Operating Characteristic Curve (RoC-AUC) results, which measure detection performance over thresholds, are reported in Table 2 for CIFAR-10 and Table 3 for ImageNet. Additional CIFAR-100 and architecture-generalization results are provided in Appendix E and Appendix L.

On CIFAR-10, RT achieves strong performance across attacks that disrupt internal representations (e.g., FGSM, CW, AutoAttack), aligning with its role in capturing large, localized deviations in intermediate layers. LT performs particularly well against attacks like PGD and Square, which introduce smaller intermediate distortion but induce instability at the output layer. Notably, the fused RLT score achieves either the best or second-best AUC across most of the attacks, validating the complementary nature of RT and LT. Despite relying solely on the internal signals of the target classifier and requiring no adversarial examples or external models, RLT is competitive with strong baselines, including BEYOND, which uses large pre-trained SSL representations.

On ImageNet, we observe similar trends. RT continues to perform well under strong perturbations such as PGD, while LT maintains robust accuracy under less structured noise like FGSM. The combined RLT score again leads to superior or competitive detection performance. Unlike several baselines that degrade

Table 7: Evaluation of RA (%) on end-to-end gradient-free SimBA attack with 1000 steps under perturbation $\epsilon = 8/255$ (ResNet110, CIFAR10).

| Defense | RA@FPR5% | RA@FPR50% |
|---|---|---|
| None | | 5.54 |
| Ours (RLT) | 97.62 | 97.85 |

Figure 3: Empirical validation of the **A Few Large Shifts Assumption** using CIFAR-10 and ResNet-110. We plot layer-wise error distributions $\sigma(e_k)$ under different attack methods.

substantially in the large-scale setting due to increased model capacity or overfitting to specific attacks, our method remains stable without requiring additional training resources or architecture-specific tuning.

## 3.3 Detection Performance under Adaptive Attacks

To reduce the risk of gradient-obfuscation artifacts and provide end-to-end gradient flow through our detection framework, we apply the Backward Pass Differentiable Approximation (BPDA) Athalye et al. (2018) to components that may otherwise block gradients, such as the quantile module.

We first evaluate robustness under the *Orthogonal-PGD* adaptive attack following BEYOND Zhiyuan et al. (2024), where the adversary has full knowledge of both the classifier and our detection mechanism and explicitly optimizes to induce misclassification while suppressing detection. Table 4 reports robust accuracy (RA) at two $L_\infty$ budgets (0.01 and 8/255) and operating points (FPR=5% and FPR=50%). While BEYOND benefits from additional SSL components, our method attains competitive or superior RA using only model-internal signals and benign calibration, without adversarial data or external encoders. These results indicate that adaptive optimization remains constrained in this setting; we interpret the RT/LT interaction as an empirical complementarity rather than an unconditional proof of adaptive robustness.

We further evaluate fully end-to-end PGD attacks on the fused RLT score using the untargeted minimization objective $\min_{\|\delta\|_\infty \le \epsilon} \{-\mathcal{L}_{cls}(x + \delta, y) + \lambda \cdot \text{RLT}(x + \delta)\}$. Equivalently, the first term maximizes classification loss to induce misclassification, while the second term penalizes high detector scores to encourage evasion. Table 5 shows that, although RA decreases under this strong adaptive threat, our method consistently outperforms the reported adaptive-attack results of Mao et al. and BEYOND at FPR=5% across all tested $\lambda$ values. When combined with an ATC (Table 6), the system remains substantially more robust, achieving over 93.3% RA at FPR=5%.

As an additional check against gradient-masking explanations, we evaluate a query-only, gradient-free attack (SimBA) Guo et al. (2019) with 1,000 queries. Table 7 shows that although SimBA reduces the undefended classifier's RA to 5.54%, our RLT detector restores RA to 97.62% at 5% FPR. This result supports the claim that the observed gains are not solely due to first-order gradient artifacts, while stronger future adaptive evaluations remain valuable.

## 3.4 Empirical Evaluation of the Proposed Assumption

To validate our **A Few Large Shifts Assumption**, we first measured layer-wise reconstruction error distributions $\sigma(e_k)$ on a ResNet-110 model trained on CIFAR-10 under various attacks: FGSM, PGD, CW,

Table 8: Comparison of implementation costs in terms of FLOPs, parameters, and model size overhead when the target model is ResNet110 with the CIFAR-10 dataset.

| Method | FLOPs (G) | Params (M) | Model Size Overhead ($\times$) |
| --- | --- | --- | --- |
| Mao Mao et al. (2021) | 5.25 | 38.12 | 22.02 |
| LNG Abusnaina et al. (2021) | **0.286** | *8.33* | *4.81* |
| BEYOND Zhiyuan et al. (2024) | 0.715 | 20.62 | 11.91 |
| **Ours (RLT)** | *0.491* | **2.59** | **1.49** |

AutoAttack, and Square Attack. As shown in Figure 3, benign inputs exhibit relatively flat error profiles across layers, while several attacks produce sharp peaks at specific layers, indicating localized shifts in internal representations. Additional substantial validations are presented in Appendix L. Across 12 dataset/model combinations and seven perturbation types, benign perturbations (random crop, horizontal flip, and random noise) have low layer-wise Lipschitz violation rates (3.57% on average), whereas adversarial perturbations have high violation rates (87.51% on average). The expanded LT analysis further shows that CW and AutoAttack examples with low RT/layer-wise violation can still have high LT scores, supporting the need to use RT and LT jointly. We further analyze calibration stability and RT/LT failure modes in Appendix J. This additional analysis shows that representative benign calibration sets of 500–1,000 samples already recover near-nominal 5% held-out FPR under the expanded validation setting, while deliberately narrow one-transformation calibration sets over-flag other benign shifts. We therefore require benign calibration to be representative of the intended deployment distribution and recalibrated when the target classifier or data distribution changes.

### 3.5 Implementation Costs

We evaluate the computational efficiency of our RLT method relative to existing baselines under the same configuration as in Table 2. As shown in Table 8, RLT introduces low floating-point operations (FLOPs), parameters, and model size overhead, offering a lightweight, plug-in solution without retraining the target classifier or using external pretrained models. It is significantly more efficient than SSL-based (Mao et al. Mao et al. (2021)) and graph-based (LNG Abusnaina et al. (2021)) detectors, making it suitable for resource-constrained deployments when the auxiliary RT/LT calibration cost is acceptable. A detailed analysis of this detection overhead across various architectures is provided in Appendix G.

## 4 Conclusion

In this work, we introduced a framework for adversarial example detection that leverages layer-wise inconsistencies within deep neural networks. Motivated by our proposed **A Few Large Shifts Assumption**–which posits that adversarial perturbations often cause large, localized violations of *layer-wise Lipschitz continuity*–we developed two complementary detection strategies: RT and LT. Extensive evaluations on CIFAR-10, CIFAR-100, and ImageNet demonstrate that our combined approach, RLT, achieves strong detection performance against a wide range of threats, including adaptive attacks. Our method operates efficiently without relying on external models or extensive augmentation, and our system-level analysis provides a practical method for selecting a detection threshold with a lower bound on accuracy under explicitly stated assumptions, highlighting its suitability as a plug-in detector rather than a standalone robustness guarantee.

**Limitations and Future Work.** Despite these results, the framework's effectiveness relies on an assumption grounded in empirical observations of current attacks and thus cannot guarantee defense against all possible future attacks. Key directions for future work therefore include deepening the theoretical understanding of *layer-wise Lipschitz continuity* to determine if the "A Few Large Shifts" principle is a fundamental property of adversarial examples, and developing more sophisticated methods for analyzing the challenging logit space. This exploration could lead to provably robust defenses. The calibration analysis also clarifies a deployment requirement: thresholds and empirical CDFs should be calibrated with representative benign data for each target model and deployment distribution. If the classifier, dataset, preprocessing pipeline, or benign distribution changes, recalibration is required. We have validated a modern ViT-B/16 target in addition to several CNN families, but target models trained with substantially different objectives, such as DINO- or MAE-style self-supervised and masked-image-modeling objectives, remain important future evaluations.

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

## Appendix

## A   Related Work and Background

**Input Transformation-Based Detectors.**   A well-established line of work detects adversarial examples by applying simple input transformations and monitoring the model's response. For instance, Xu et al. (2017) proposed *feature squeezing*, which reduces input precision or applies spatial smoothing to suppress adversarial noise. Liang et al. (2018) used quantization and denoising techniques, while other works explore random crops or region replacements. These preprocessing-based detectors are lightweight and model-agnostic but assuming that perturbations are fragile to such transformations. Their effectiveness often hinges on tuning a detection threshold, and their robustness degrades against adaptive attacks that are explicitly trained to remain invariant under the applied transformations. Coarse transformations may also degrade clean-data performance, increasing false positives.

**Feature Statistics-Based Detectors.**   Another category of detectors analyzes statistical anomalies in the network's hidden representations. Ma et al. (2018) proposed Local Intrinsic Dimensionality (LID) to measure how local feature neighborhoods expand under adversarial perturbations. Feinman et al. (2017) leveraged kernel density estimation and uncertainty measures, while Lee et al. (2018) introduced a Mahalanobis distance-based detector by modeling class-conditional distributions in intermediate layers. These approaches rely on hand-crafted features from latent activations and often require storing high-dimensional embeddings or computing pairwise distances, which can be computationally intensive. Moreover, many methods are vulnerable to adaptive attacks that mimic the distributional properties of benign inputs.

**Model Behavior-Based Detectors.**   Some methods probe the internal behavior of the model itself to identify inconsistencies caused by adversarial perturbations. Lu et al. (2017) encoded final-layer activations into binary vectors for SVM-based detection, while Metzen et al. (2017) appended auxiliary classifiers to intermediate layers. Others like Ma & Liu (2019) and Carrara et al. (2018) focused on monitoring neuron activation paths or ensemble agreement. These methods often require additional training, architectural modifications, or model ensembles, increasing overhead and complicating deployment. Many also depend on adversarial examples for supervision, which limits their generalizability to unseen attacks.

**Self-Supervised and Consistency-Based Detectors.**   Recent approaches have explored feature consistency using self-supervised learning (SSL) models. For example, BEYOND Zhiyuan et al. (2024) detects adversarial inputs by measuring feature stability across augmentations in a large SSL encoder. These methods eliminate the need for adversarial examples and external graph structures but introduce high computational costs due to reliance on large pretrained models, which may be significantly larger than the target classifier itself (e.g., 22× larger parameters). Additionally, they often require access to domain-specific SSL models and retraining or fine-tuning of auxiliary heads.

**Our Approach.**   In contrast to the methods above, our framework introduces a fundamentally different detection paradigm. Conceptualizing the network as a cylinder where each layer is a horizontal slice, prior work can be seen as performing a "horizontal" analysis: comparing a sample's features against a reference set or augmentations within a single layer. We propose a "vertical" analysis that is entirely self-referential. Our method scrutinizes the consistency of feature transformations between the network's own layers for a single input, eliminating the need for the external reference data, heavy augmentations, or complex data structures common in horizontal methods. This approach is grounded in our **A Few Large Shifts Assumption**, which posits that adversarial perturbations cause large, localized deviations between a few critical layers, making these vertical inconsistencies a detectable signal.

## B   Theoretical Analysis

We provide a sufficient-condition analysis for our proposed detection framework, which includes RT, LT, and the fused score RLT. The results below should be read as explanatory analysis under explicitly stated assumptions, not as an unconditional proof that all adversarial examples must be detected. For each score, we analyze how the metric behaves when the corresponding layer-wise Lipschitz or logit-instability assumptions hold, and we pair this analysis with the empirical validation in Appendix L.

## B.1 Connection to Lipschitz Continuity

Our detection framework can be formally understood as an empirical method for identifying violations of Lipschitz continuity across localized layers, we term the violation of *layer-wise Lipschitz continuity*. Let $f_k$ be the function corresponding to the $k$-th layer of the network, such that $\mathbf{z}_k = f_k(\mathbf{z}_{k-1})$. The local Lipschitz constant $L_k$ at an input $\mathbf{z}_{k-1}$ is the smallest value such that:

$$\|f_k(\mathbf{z}'_{k-1}) - f_k(\mathbf{z}_{k-1})\| \leq L_k \|\mathbf{z}'_{k-1} - \mathbf{z}_{k-1}\| \tag{7}$$

for all $\mathbf{z}'_{k-1}$ in a neighborhood of $\mathbf{z}_{k-1}$. Our **A Few Large Shifts Assumption** posits that for an adversarial input $x^{\mathrm{adv}}$, the ratio $\frac{\|\mathbf{z}_{k+1}(x^{\mathrm{adv}}) - \mathbf{z}_{k+1}(x)\|_2}{\|\mathbf{z}_k(x^{\mathrm{adv}}) - \mathbf{z}_k(x)\|_2}$ becomes very large for a small subset of layers $\mathcal{T}$. This ratio serves as an empirical estimate of the local Lipschitz constant for the layer transformation $f_{k+1}$.

Our Recovery Testing (RT) score is designed to detect these violations. A large reconstruction error, $e_k(x)$, signals a significant deviation between the expected and actual feature transformations, which is a manifestation of this high layer-wise Lipschitz constant. By calibrating the expected error distributions on benign data—where we assume layer-wise Lipschitz constants are small and stable—RT effectively identifies inputs that cause these localized instabilities.

## B.2 Justification for Layer-wise Reconstruction Error in RT

This section analyzes why using the reconstruction error of intermediate features—as done in RT—produces a separable distribution between benign and adversarial inputs. Specifically, we show that benign inputs produce consistently low recovery residuals, while adversarial perturbations, though small in input space, induce disproportionately large deviations in the representation space at select layers, leading to higher RT scores.

**Assumption 2** (Approximate Invertibility). *For each intermediate layer $i \in \{k_{RT}, \ldots, L-1\}$, there exists a well-trained inverse function $R^{(L \to i)}$ such that, for benign inputs $x$,*

$$\|z_i(x) - R^{(L \to i)}(z_L(x))\| \leq \varepsilon,$$

*for some small constant $\varepsilon > 0$.*

**Assumption 3** (Flatness of the Inverse Function). *The inverse function $R^{(L \to i)}$ is Lipschitz-smooth with constant $\alpha \ll 1$. That is, for all small perturbations $\delta$,*

$$\|R^{(L \to i)}(z_L(x + \delta)) - R^{(L \to i)}(z_L(x))\| \leq \alpha \cdot \|z_L(x + \delta) - z_L(x)\|.$$

**Assumption 4** (Sub-Gaussian Layer Perturbations). *Let $\delta$ be an adversarial perturbation. Then the induced change in intermediate features is sub-Gaussian:*

$$\|\nabla z_i(x) \cdot \delta\| \sim SubG(\mu, \sigma^2),$$

*i.e., for all $t > 0$,*

$$\Pr\left(\|\nabla z_i(x) \cdot \delta\| \leq \mu - t\right) \leq \exp\left(-\frac{t^2}{2\sigma^2}\right).$$

**Theorem 1** (RT Detects Adversarial Residuals). *Under Assumption 1, we assume that $\mathcal{T} \subseteq \{f_1, \ldots, f_{L-1}\}$ contains one or more intermediate layers where adversarial perturbations induce disproportionately large shifts. These shifts lead to elevated reconstruction residuals, which RT is designed to detect. Let $x^{\mathrm{adv}} = x + \delta$ be an adversarial example, and suppose Assumption 2, 3, and 4 hold. Then, with probability at least $1 - \eta$ for small $\eta$, the following inequality holds:*

$$\|z_i(x^{\mathrm{adv}}) - R^{(L \to i)}(z_L(x^{\mathrm{adv}}))\|^2 > \|z_i(x) - R^{(L \to i)}(z_L(x))\|^2.$$

(Proof: Appendix C.1)

### B.3 Justification for Ratio-based Logit-layer Deviation in LT

In this section, we justify the use of the LT score, which captures how the logit-layer output changes under augmentation relative to changes in intermediate-layer features. We show that benign inputs maintain logit consistency under mild perturbations, while adversarial examples—particularly those crafted to flip decisions—exhibit exaggerated logit volatility, causing the LT score to grow disproportionately.

**Assumption 5** (Benign Augmentation Stability). *For any benign input $x$, and a mild transformation $W^{(g)}$, the resulting features satisfy*

$$\|z^{(g)} - z\| \leq \eta, \quad \|\ell^{(g)} - \ell\| \leq \alpha(\eta),$$

*for small $\eta > 0$ and monotonically increasing $\alpha(\eta) \ll 1$.*

**Assumption 6** (Recovery Test Evasion). *Let $x^{\mathrm{adv}} = x + \delta$ be an adversarial input that satisfies Assumption 1 with $\mathcal{T} = \{f_{logit}\}$, i.e., the perturbation induces a large shift only at the logit layer, while intermediate representations remain largely consistent with those of benign inputs.*

*Then the residuals measured by RT remain low:*

$$\|z_i(x^{\mathrm{adv}}) - R^{(L \to i)}(z_L(x^{\mathrm{adv}}))\|^2 \lesssim \|z_i(x) - R^{(L \to i)}(z_L(x))\|^2,$$

*yet flips the final prediction:*

$$\arg\max \ell(x^{\mathrm{adv}}) \neq \arg\max \ell(x).$$

*Moreover, augmentations preserve feature drift:*

$$\|z^{(g),adv} - z^{adv}\| \approx \|z^{(g)} - z\|.$$

**Theorem 2** (Logit Instability under Augmentation). *Under Assumption 6, let $\hat{y} = \arg\max_c \ell_c(x^{\mathrm{adv}})$ and $\hat{y}_g = \arg\max_c \ell_c(x^{(g),\mathrm{adv}})$. The adversarial logit output is unstable under small augmentation when*

$$\|\mathbf{o}_{\hat{y}} - \mathbf{o}_{\hat{y}_g}\| > 0.$$

(Proof: Appendix C.2)

**Theorem 3** (Amplified Logit Sensitivity). *Under Assumption 5 and Assumption 6, the logit sensitivity of adversarial inputs satisfies:*

$$\frac{\|\ell^{(g),adv} - \ell^{adv}\|}{\|z^{(g),adv} - z^{adv}\|} > \frac{\|\ell^{(g)} - \ell\|}{\|z^{(g)} - z\|}.$$

(Proof: Appendix C.3)

### B.4 Quantile-normalized RT + LT Provides Jointly Separable Score in RLT

We now provide a theoretical justification for the fused detection score $RLT(x) = RT_{\mathrm{norm}}^2 + LT_{\mathrm{norm}}^2$. Since RT and LT each capture different types of adversarial signatures (internal layers misalignment vs. logit instability), combining them creates a more robust metric. By applying quantile normalization, we map both scores into a common distributional space, ensuring fair fusion. We then show that the fused score statistically separates adversarial inputs even when only one metric is significantly perturbed.

**Assumption 7** (Quantile-normalized RT and LT). *Let $\hat{\mathcal{F}}_{RT}, \hat{\mathcal{F}}_{LT}$ be empirical CDFs computed on benign RT and LT scores. Define:*

$$RT_{norm}(x) = \Phi^{-1}(\hat{\mathcal{F}}_{RT}(RT(x))), \quad LT_{norm}(x) = \Phi^{-1}(\hat{\mathcal{F}}_{LT}(LT(x))),$$

*where $\Phi^{-1}$ is the standard normal quantile function.*

**Assumption 8** (Adversarial Score Margin). *There exists $\gamma > 0$ such that for adversarial $x$, at least one normalized score satisfies:*

$$|RT_{norm}(x)| > \gamma \quad or \quad |LT_{norm}(x)| > \gamma.$$

**Theorem 4** (RLT Separates Adversaries). *Define the fused score:*

$$RLT(x) = RT_{norm}(x)^2 + LT_{norm}(x)^2.$$

*Then, under Assumption 7 and Assumption 8,*

$$\mathbb{E}[RLT(x)] = 2 \quad for \ benign \ x, \quad and \quad RLT(x^{adv}) > \gamma^2.$$

(Proof : Appendix C.4)

### B.5 Robustness to Adaptive Attacks

In this section, we provide a theoretical perspective on the robustness of our detection framework under adaptive adversaries. An adaptive attack refers to a threat model where the adversary has full knowledge of both the classifier and the detection mechanism, and explicitly optimizes its objective to evade detection. We show that the design of our RT and LT metrics inherently introduces conflicting optimization gradients, which hinder the adversary's ability to jointly suppress both detection scores.

#### B.5.1 Adaptive Attack Objective

For an untargeted white-box adversary, the attack must both induce misclassification and evade the detector. We therefore minimize the following constrained objective:

$$\min_{\|\delta\| \leq \epsilon} \left[ -\mathcal{L}_{cls}(x + \delta, y) + \beta_1 RT(x + \delta) + \beta_2 LT(x + \delta) \right],$$

where the negative classification loss term is equivalent to maximizing cross-entropy for misclassification, and $\beta_1, \beta_2 \geq 0$ trade off detector suppression. For a targeted attack toward $y_t$, the classification term becomes $\mathcal{L}_{cls}(x + \delta, y_t)$. The fused-score attack used in Table 5 sets $\beta_1, \beta_2$ through the quantile-normalized RLT score and sweeps the scalar detector weight $\lambda$. While the classification term drives misclassification, the additional RT and LT terms encourage the adversary to remain internally and logit-wise consistent. We empirically observe in Table 13 and Figure 7 that attacks with weak layer-wise violation can still be separated by LT, supporting the practical complementarity of the two objectives.

#### B.5.2 Conflicting Gradient Effects

The RT score is defined as:

$$RT(x) \propto \|z_i(x) - R^{(L \to i)}(z_L(x))\|^2,$$

which penalizes deviations from the inverse-mapped intermediate representations. Minimizing this score encourages the adversary to maintain stable internal features consistent with benign patterns. In contrast, the LT score is defined as:

$$LT(x) \propto \frac{\mathcal{H}(\sigma(\ell(x))) \cdot \Delta \ell^{(g)}(x)}{\Delta z^{(g)}(x)},$$

where $\Delta \ell^{(g)}(x)$ measures logit deviation across augmentations, and $\Delta z^{(g)}(x)$ captures the corresponding feature drift. Minimizing LT encourages logit stability while allowing some augmentation-induced feature variability. These two goals can compete. Enforcing small residuals in intermediate layers (RT) limits the allowable variation in augmented features, which may inflate LT. Conversely, promoting augmentation-invariant logits (LT) can introduce instability in internal features, increasing RT. We therefore present the "conflicting gradients" explanation as an empirically supported mechanism rather than a formal guarantee. In practice, we observe that attempting to suppress one score often exacerbates the other, making it difficult to minimize the fused detection score $RLT(x) = RT^2_{norm}(x) + LT^2_{norm}(x)$ in the evaluated attacks.

## C Proofs

### C.1 Theorem 1

*Proof.* For the benign input $x$, Assumption 2 guarantees:

$$\|z_i(x) - R^{(L \to i)}(z_L(x))\| \leq \varepsilon.$$

Now consider the adversarial input $x^{\mathrm{adv}} = x + \delta$. Using first-order Taylor expansions:

$$z_i(x^{\mathrm{adv}}) = z_i(x) + \nabla z_i(x) \cdot \delta + o(\|\delta\|),$$

$$z_L(x^{\mathrm{adv}}) = z_L(x) + \nabla z_L(x) \cdot \delta + o(\|\delta\|).$$

Next, apply the inverse recovery map:

$$R^{(L \to i)}(z_L(x^{\mathrm{adv}})) = R^{(L \to i)}(z_L(x)) + \nabla R^{(L \to i)}(z_L(x)) \cdot (z_L(x^{\mathrm{adv}}) - z_L(x)) + o(\|\delta\|).$$

Define:

$$\Delta_z := \nabla z_i(x) \cdot \delta, \quad \Delta_r := \nabla R^{(L \to i)}(z_L(x)) \cdot \nabla z_L(x) \cdot \delta.$$

Then:

$$z_i(x^{\mathrm{adv}}) - R^{(L \to i)}(z_L(x^{\mathrm{adv}})) \approx (z_i(x) - R^{(L \to i)}(z_L(x))) + (\Delta_z - \Delta_r).$$

By reverse triangle inequality:

$$\|z_i(x^{\mathrm{adv}}) - R^{(L \to i)}(z_L(x^{\mathrm{adv}}))\| \geq \|\Delta_z\| - \|z_i(x) - R^{(L \to i)}(z_L(x))\| - \|\Delta_r\|.$$

From Assumption 2 and Assumption 3:

$$\|z_i(x) - R^{(L \to i)}(z_L(x))\| \leq \varepsilon, \quad \|\Delta_r\| \leq \alpha \cdot \|\nabla z_L(x) \cdot \delta\|.$$

Assuming $\mu > 2(\varepsilon + \alpha \cdot \|\nabla z_L(x) \cdot \delta\|)$, Assumption 4 implies that with probability at least $1 - \eta$,

$$\|\Delta_z\| \geq \mu > 2\varepsilon + \|\Delta_r\|,$$

where $\eta \leq \exp\left(-\frac{(\mu - 2(\varepsilon + \|\Delta_r\|))^2}{2\sigma^2}\right)$. Then:

$$\|z_i(x^{\mathrm{adv}}) - R^{(L \to i)}(z_L(x^{\mathrm{adv}}))\| \geq \varepsilon.$$

Squaring both sides proves the theorem. $\qquad\square$

### C.2 Theorem 2

*Proof.* Given the prediction is flipped but the intermediate features remain close to those of $x$, the adversarial logit lies near a decision boundary. Thus, even a mild augmentation $x^{(g),\mathrm{adv}}$ can shift the logits across the boundary, changing the predicted class. $\qquad\square$

### C.3 Theorem 3

*Proof.* Assume the contrary. Then adversarial sensitivity is less than benign. Given $\|z^{(g),\mathrm{adv}} - z^{\mathrm{adv}}\| \approx \|z^{(g)} - z\|$, this implies:

$$\|\ell^{(g),\mathrm{adv}} - \ell^{\mathrm{adv}}\| \leq \|\ell^{(g)} - \ell\| \leq \alpha(\eta).$$

Hence, $\ell^{(g),\mathrm{adv}} \approx \ell^{\mathrm{adv}}$, contradicting Theorem 2. Thus the adversarial logit must be more sensitive to benign augmentation than the original. $\qquad\square$

**C.4 Theorem 4**

*Proof.* Since both normalized scores follow $\mathcal{N}(0,1)$, the expected value of their squared sum under benign data is:

$$\mathbb{E}[RLT(x)] = \mathbb{E}[RT^2_{\text{norm}}] + \mathbb{E}[LT^2_{\text{norm}}] = 1 + 1 = 2.$$

For adversarial inputs, Assumption 8 ensures at least one squared score exceeds $\gamma^2$. Thus,

$$RLT(x^{\text{adv}}) > \gamma^2.$$

$\square$

# D   Ablation Study

All ablation studies were conducted using the CIFAR-10 dataset with ResNet110. We systematically vary the key hyperparameters and architectural choices to assess their impact on detection performance.

## D.1   Size of recover module

We evaluate the impact of varying the depth and dimensionality of the recover modules on RT detection performance. As we reduce the size of the recover modules in terms of both depth and dimensionality, we observe slightly improved performance as shown in Table 9. This result aligns well with Assumption 3, which posits that a flatter or smoother recovery module—achieved by reducing complexity—results in improved detection due to closer alignment with the assumption of stable layer-wise reconstruction under benign conditions. However, the overall performance differences are relatively minor, demonstrating robustness to the choice of recover module size.

Table 9: RT test AUC (%) differences when varying depth and dimensionality of recover modules, compared to depth = 5 and dimensionality = 512.

| Depth | FGSM | PGD | CW | AutoAttack | Square | Avg. |
|---|---|---|---|---|---|---|
| 2 | +0.55 | -2.03 | +0.16 | 0.00 | +10.96 | +1.93 |
| 3 | +0.15 | +0.15 | +0.08 | 0.00 | +4.25 | +0.92 |
| 4 | -0.01 | +0.01 | +0.04 | 0.00 | -0.50 | -0.09 |
| 5 | 0.00 | 0.00 | 0.00 | 0.00 | 0.00 | 0.00 |
| **Dimensionality** | **FGSM** | **PGD** | **CW** | **AutoAttack** | **Square** | **Avg.** |
| 64 | -0.50 | -0.10 | -0.40 | 0.00 | +8.34 | +1.47 |
| 128 | -0.54 | -0.04 | -0.20 | 0.00 | +4.13 | +0.67 |
| 256 | -0.18 | -0.18 | -0.03 | 0.00 | +0.65 | +0.13 |
| 512 | 0.00 | 0.00 | 0.00 | 0.00 | 0.00 | 0.00 |

## D.2   Number of learnable augmentations $G$

We investigate how varying the number of learnable augmentation matrices ($G$) impacts LT detection performance. As depicted in Table 10, the detection performance exhibits minimal sensitivity to the number of augmentations used. Even with a significantly smaller number of augmentations ($G < 4$), LT continues to perform robustly. This indicates our LT method effectively quantifies logit-layer perturbations without relying on extensive augmentation, in contrast to previous methods such as BEYOND Zhiyuan et al. (2024), which required up to 50 augmentations. The minimal requirement of augmentations highlights our method's computational efficiency and practical deployability.

Table 10: LT test AUC (%) differences when varying the number of augmentation matrices ($G$), compared to $G = 4$.

| $G$ | FGSM | PGD | CW | AutoAttack | Square | Avg. |
|---|---|---|---|---|---|---|
| 1 | +0.01 | -0.03 | -0.02 | -0.02 | +0.01 | -0.01 |
| 2 | -0.04 | +0.08 | -0.14 | +0.37 | +0.03 | +0.06 |
| 3 | 0.00 | +0.20 | -0.05 | +0.46 | +0.02 | +0.12 |
| 4 | 0.00 | 0.00 | 0.00 | 0.00 | 0.00 | 0.00 |

### D.3 Choice of $k_{RT}$ and $k_{LT}$

We examine how choosing identical values for hyperparameters $k_{RT}$ and $k_{LT}$ affects combined RLT performance relative to optimal, independent selection of these parameters. Results presented in Table 11 indicate a maximum performance decrease of only 1.33% when setting $k = k_{RT} = k_{LT}$ uniformly, as opposed to independently optimizing each hyperparameter. These findings highlight the practical robustness of our detection framework to hyperparameter selection, emphasizing that fine-grained tuning of $k_{RT}$ and $k_{LT}$ is unnecessary for achieving high detection accuracy, greatly simplifying deployment.

Table 11: RLT test AUC (%) differences for varying values of $k = k_{RT} = k_{LT}$ compared to optimal separate selection.

| $k = k_{RT} = k_{LT}$ | FGSM | PGD | CW | AutoAttack | Square | Avg. |
|---|---|---|---|---|---|---|
| 1 | -1.59 | -3.45 | -0.82 | -0.03 | -0.77 | -1.33 |
| 10 | -1.37 | -3.55 | -0.49 | -0.04 | -0.86 | -1.26 |
| 15 | -1.03 | -3.41 | -0.35 | -0.04 | -1.02 | -1.17 |
| 20 | -1.25 | -3.16 | -0.33 | -0.04 | -0.63 | -1.08 |
| 25 | -1.37 | -2.51 | -0.37 | -0.03 | -1.05 | -1.06 |
| 30 | -1.67 | -1.33 | -0.39 | -0.01 | -0.77 | -0.83 |
| $k_{RT} \neq k_{LT}$ (Optimal) | 0.00 | 0.00 | 0.00 | 0.00 | 0.00 | 0.00 |

### D.4 Single objective adaptive attacks

To highlight the inherent robustness of our framework, we begin with an ablation study under a simplified setting. Specifically, we consider cases where both the attacker and the defender rely on a single detection objective—either RT or LT. As shown in Table 12, this restriction leads to substantial performance degradation compared to the full defense, where both RT and LT are jointly employed for attack and detection. These results demonstrate that neither component alone is sufficient.

Table 12: Single Objective Orthogonal-PGD Adaptive Attack on $L_\infty = 8/255$ using CIFAR-10 and ResNet110.

| Removed Objective | RA@FPR5% | RA@FPR50% |
|---|---|---|
| None | **33.70%** | **80.77%** |
| RT | 17.11% | 55.30% |
| LT | 17.20% | 56.17% |

To provide a more comprehensive analysis, we conducted an additional experiment where the attacker's objective may differ from the defender's measurement score. As shown in Table 13, an adversary attacking the full RLT objective achieves the lowest average Robust Accuracy (RA) across all defender configurations (64.85%). This demonstrates that the fused RLT framework poses a significant challenge for an adversary. The most effective evasion strategy is to target the combined RLT score directly, as attacking the individual RT or LT components results in a lower expected attack success rate.

Table 13: RA@FPR50% (%) under mismatched end-to-end PGD attack and defense objectives ($\epsilon = 8/255$, ResNet110, CIFAR-10).

| Measured With ↓ / Attacked Measure → | RLT | RT | LT |
|---|---|---|---|
| **RLT** | $\mathbf{68.53}_{\pm\mathbf{2.21}}$ | $72.06_{\pm 4.05}$ | $73.05_{\pm 2.99}$ |
| **RT** | $78.75_{\pm 1.53}$ | $\mathbf{75.56}_{\pm\mathbf{1.30}}$ | $81.58_{\pm 1.73}$ |
| **LT** | $47.28_{\pm 4.43}$ | $64.49_{\pm 8.12}$ | $\mathbf{46.48}_{\pm\mathbf{3.67}}$ |
| **Average** | **64.85** | **70.70** | **67.04** |

This outcome empirically substantiates our theoretical analysis: adaptive attacks face inherent difficulties due to conflicting gradient directions induced by RT and LT, which makes attack algorithms difficult to find $\mathcal{T}$. When an adversary focuses only on evading RT, they tend to create perturbations that shift the logit layer, which are then caught by LT. Conversely, attacks targeting only LT tend to disrupt intermediate features, which are then caught by RT. These distinct perturbation strategies confirm our assumption that RT and LT effectively impose opposing constraints on adaptive adversaries, making the combined RLT score a significantly more robust defense.

### D.5 Ablation on perturbation budget sensitivity of each testing measure

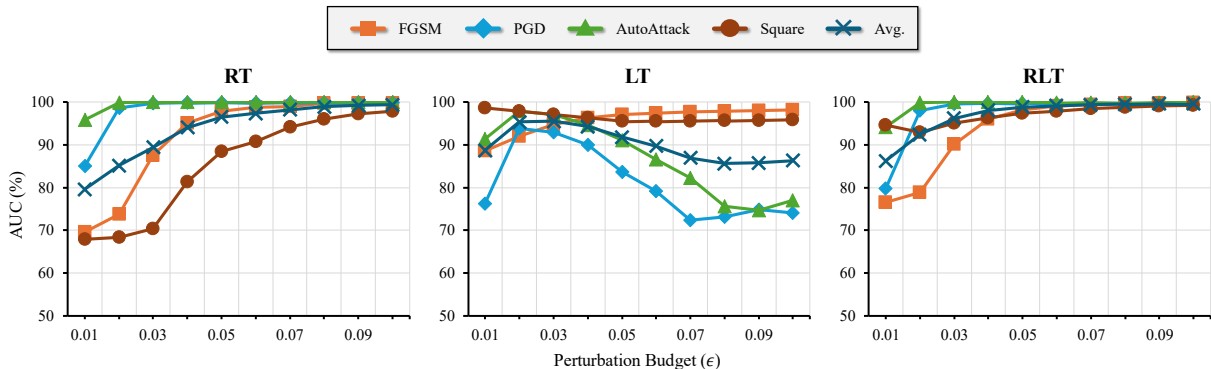

Figure 4: Ablation study showing AUC (%) of RT, LT, and RLT under varying perturbation budgets ($\epsilon \in \{0.01, 0.02, \cdots, 0.1\}$) across multiple standard attack types (FGSM, PGD, AutoAttack, Square). Evaluated on CIFAR-10 using a ResNet-110 classifier and detection models trained with fewer epochs for efficiency.

To analyze the contribution of each detection component across adversarial strengths, we conduct an ablation study varying the perturbation budget $\epsilon$ for common standard (non-adaptive) attacks, including FGSM, PGD, AutoAttack, and Square. We report AUC detection performance for RT, LT, and RLT. All experiments are performed on the CIFAR-10 dataset using a pretrained ResNet-110 classifier. Detection models are trained with a reduced number of epochs to simulate lightweight deployment.

As visualized in Figure 4, detection performance consistently improves as the adversary's perturbation budget increases. This trend reflects our framework's strength in leveraging internal inconsistencies that become more pronounced under stronger attacks. Unlike previous methods, such as BEYOND Zhiyuan et al. (2024), which tend to suffer degraded performance at higher perturbation levels due to their reliance on final feature stability, our method benefits from capturing the greater representational disruption induced across the layers by adversaries at higher budgets.

We also observe that as the budget increases, the targeted layers by adversarial perturbations tend to shift from the logit layer to deeper intermediate layers. This is evidenced by the decline in LT's AUC and the concurrent improvement in RT's performance. We interpret this behavior as a reflection of the

limited perturbation capacity of each layer, under the assumption that each layer contains only a finite set of vulnerable manifolds. As the perturbation budget grows, the adversary exhausts the capacity of the logit layer and is forced to exploit deeper, intermediate representations. Despite this divergence, RLT– which combines both signals–demonstrates a smooth and consistent increase in performance, validating its robustness and complementary design.

This supports our **A Few Large Shifts Assumption**, reinforcing that adversarial perturbations typically cause disproportionately large disruptions in a small subset of layers, which become more detectable as the attack budget increases. The results further demonstrate that combining RT and LT provides stable and effective detection across a wide perturbation spectrum.

### D.6 Contribution of each term in testing measures

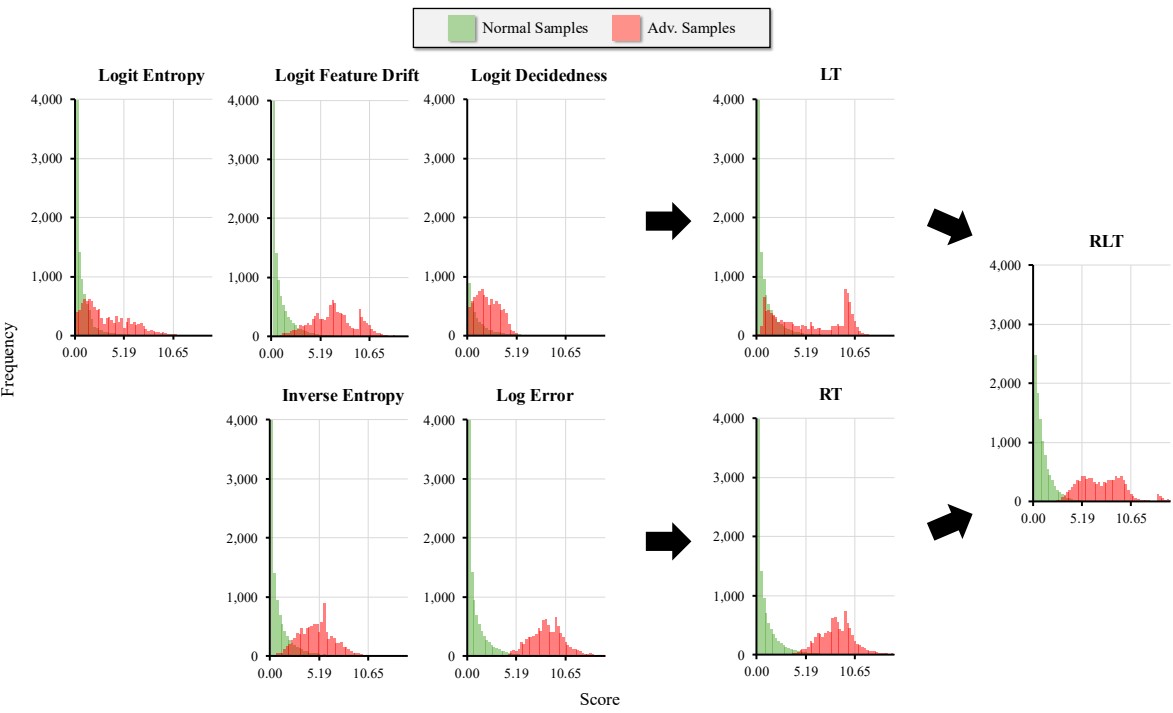

Figure 5: Empirical distributions of individual terms used in our detection metrics, evaluated on CIFAR-10 with a ResNet-110 model under FGSM attack ($\epsilon = 0.05$). Scores are squared and normalized using quantile normalization based on clean test samples.

To better understand the role of each component in our proposed detection metrics, we perform an ablation study by individually removing key terms from the Logit-layer Testing and Recovery Testing scores. We evaluate the resulting change in detection performance using AUC scores under five adversarial attack types– FGSM, PGD, CW, AutoAttack, and Square–on CIFAR-10 with a ResNet-110 classifier. Table 14 summarizes the results along with the average performance drop caused by each ablation.

We observe that LT is particularly sensitive to the removal of the logit entropy and feature drift terms. Excluding either leads to a substantial degradation in performance across all attacks, suggesting that these terms are central to LT's ability to quantify abnormal sensitivity in the output layer relative to internal representation changes. The entropy term, in particular, governs the calibration of the detector based on output uncertainty and appears essential for capturing confidence shifts introduced by adversarial perturbations. The feature drift term complements this by providing a baseline of expected internal variation under benign

Table 14: Contribution of each term in the LT and RT metrics. AUC scores (in percentage) are reported on CIFAR-10 using ResNet-110 under different attacks. The last column indicates the average drop in performance from the full model.

| Removed Term | FGSM | PGD | CW | AutoAttack | Square | Average | Drop |
|---|---|---|---|---|---|---|---|
| **Logit-layer Testing (LT)** | | | | | | | |
| None (Full LT) | 97.50 | 98.61 | 97.08 | 99.60 | 97.47 | 98.05 | – |
| – Logit Decidedness | 99.51 | 94.66 | 82.97 | 96.42 | 95.91 | 93.89 | -4.16 |
| – Logit Entropy | 97.51 | 59.37 | 89.76 | 57.14 | 97.20 | 80.19 | -17.86 |
| – Feature Drift | 94.62 | 65.49 | 99.29 | 50.01 | 97.29 | 81.34 | -16.71 |
| **Recovery Testing (RT)** | | | | | | | |
| None (Full RT) | 99.93 | 96.89 | 99.90 | 99.99 | 85.38 | 96.42 | – |
| – Inverse Entropy | 98.98 | 94.35 | 99.09 | 99.95 | 84.52 | 95.38 | -1.04 |
| – Log Error | 80.34 | 97.09 | 88.95 | 99.95 | 65.68 | 86.40 | -10.02 |

augmentations. Removing the logit decidedness term results in more modest, yet consistent, performance degradation, indicating its auxiliary role in sharpening the detection boundary.

In the case of RT, the core driver of performance is the reconstruction error between predicted and actual intermediate features. Its removal yields a pronounced drop in detection accuracy, reaffirming that adversarial perturbations often manifest as sharp deviations in the internal feature manifold. In contrast, the inverse entropy term, which weights errors based on the sharpness of their distribution across layers, contributes more marginally to the overall score. Its absence slightly affects performance, implying it primarily serves as a regularizer rather than a principal signal.

These findings underscore that both RT and LT derive their strength from distinct but complementary cues–internal feature consistency and output–level robustness, respectively. While some components act as critical discriminators, others enhance calibration and stability. Together, their integration within the full detection framework leads to robust and generalizable performance across diverse attack types.

Figure 5 visually complements these results by showing the empirical distributions of each term on clean and adversarial samples. We observe that each term, when considered independently, introduces a measurable degree of separation between normal and adversarial inputs. This separation is modest for individual components but becomes substantially more pronounced when the terms are combined within LT and RT. The final integration of both methods in RLT leads to the strongest separation, illustrating the complementary nature of these features.

### D.7 Evaluation Against $L_2$-Norm Attacks

To ensure our method's effectiveness is not limited to a single threat model, we evaluated its performance against $L_2$-norm attacks, demonstrating that its detection capabilities are largely norm-agnostic.

First, to establish a challenging and comparable evaluation setting, we identified a worst-case perturbation budget for an $L_2$-PGD attack. As shown in Table 15, a budget of $\epsilon = 0.6$ was sufficient to reduce the base classifier's Robust Accuracy to 0.00%.

Using this budget, we then compared our detector's performance against both $L_\infty$ and $L_2$ attacks. The results in Table 16 show that our combined RLT detector achieved a high AUC of 98.05% against the strong $L_2$ attack, which is comparable to its performance against the $L_\infty$ attack. This confirms the robustness of our detection framework across different norm-based threat models.

Table 15: RA (%) under $L_2$-PGD attack on CIFAR-10 to determine a worst-case budget.

| $\epsilon$ | RA |
|---|---|
| 0.1 | 39.97 |
| 0.2 | 6.21 |
| 0.3 | 0.74 |
| 0.4 | 0.10 |
| 0.5 | 0.01 |
| 0.6 | 0.00 |

Table 16: AUC (%) scores comparing performance against $L_\infty$ and $L_2$ PGD attacks on CIFAR-10.

| Norm | RA | RLT | RT | LT |
|---|---|---|---|---|
| $L_\infty$ | 0.00 | **99.47** | **99.49** | 96.65 |
| $L_2$ | 0.00 | 98.05 | 95.05 | **97.16** |

## E   Generalization to CNN-based Architectures

To assess the generalizability of our detection framework across diverse neural architectures, we evaluate RT, LT, and RLT on CIFAR-100 using four distinct backbone models: ResNet-18, MobileNet-V2 x0 5, ShuffleNet-V2 x0 5, and RepVGG-a0. As shown in Table 17, our methods consistently achieve high AUC scores across all architectures and attack types (FGSM, PGD, and AutoAttack), with RLT offering the most robust overall performance.

These results demonstrate that our detection strategies—rooted in internal layer inconsistency—are not only effective on traditional architectures like ResNet and DenseNet, but also extend well to lightweight and mobile-friendly models such as MobileNet and ShuffleNet, as well as convolutional variants like RepVGG. Notably, the fused score RLT maintains average AUCs above 97% on all target models, confirming the adaptability and resilience of our approach.

Importantly, the deployment of our detectors requires minimal architectural modification and no adversarial training, making them easily integrable into a wide range of existing models. This plug-and-play nature, combined with high detection efficacy, makes our method highly suitable for practical applications where architectural diversity and efficiency are key considerations.

## F   Generalization to Transformer-based Architecture

To evaluate the generality of our detection framework beyond CNNs, we applied it to a pre-trained Vision Transformer (ViT-B/16) on CIFAR-10 and CIFAR-100. The results are presented in Table 18.

We adapted our framework to the ViT-B/16 architecture by extracting features from its constituent layers. The output of the final (12th) encoder block was designated as the source embedding ($z_L$), from which all prior layer representations were reconstructed. For the initial `conv_proj` layer, patch embeddings were spatially averaged into 14 vertical strips, with a dedicated recovery module trained for each. For each of the 11 preceding Transformer encoder blocks, the $197 \times 768$ token embedding matrices were partitioned into 6 non-overlapping feature chunks. Each chunk was then aggregated to create a feature vector for its corresponding recovery module. For LT, we used $G = 4$ augmentations. In total, this design comprises 80 lightweight recovery modules and incurs a low model size overhead of just $0.337\times$ relative to the target ViT model in terms of number of model parameters, demonstrating our approach's scalability.

As shown in the table, our method consistently outperforms the LID baseline on CIFAR-10. On CIFAR-100, it matches or exceeds LID's performance against PGD and CW attacks while remaining competitive against FGSM. These strong results, achieved with minimal overhead, confirm that our layer-wise inconsistency-

Table 17: AUC scores (%) on CIFAR-100 under three white-box attacks (FGSM, $\ell_\infty$-PGD, and AutoAttack) with the same perturbation budget as the previous evaluations.

| Variant | ResNet-18 | | | | MobileNet-V2 x0 5 | | | |
|---|---|---|---|---|---|---|---|---|
| | FGSM | PGD | AutoAttack | **Avg.** | FGSM | PGD | AutoAttack | **Avg.** |
| RT | 87.83 | 99.21 | 99.98 | 95.67 | 99.02 | 99.31 | 99.99 | 99.44 |
| LT | 97.83 | 97.53 | 99.47 | 98.28 | 94.73 | 87.50 | 91.74 | 91.33 |
| RLT | 95.61 | 98.92 | 99.97 | 98.17 | 98.73 | 98.14 | 99.98 | 98.95 |
| | **ShuffleNet-V2 x0 5** | | | | **RepVGG a0** | | | |
| RT | 98.38 | 97.35 | 99.97 | 98.57 | 91.77 | 98.94 | 99.99 | 96.90 |
| LT | 94.82 | 90.34 | 96.81 | 93.99 | 90.38 | 96.57 | 99.26 | 95.40 |
| RLT | 98.16 | 96.17 | 99.94 | 98.09 | 93.49 | 98.46 | 99.98 | 97.31 |

based detection strategy generalizes effectively to modern Transformer architectures and offers significant potential for further performance tuning.

These ViT-B/16 results address architecture generality beyond classical CNNs, while the CNN-family results in Appendix E cover ResNet, DenseNet, MobileNet, ShuffleNet, and RepVGG-style models. We do not interpret these experiments as proving architecture- or training-objective-universal behavior. In particular, target models trained with self-supervised or masked-image-modeling objectives such as DINO or MAE may organize intermediate representations differently, and should be recalibrated and evaluated separately. We now state this as an explicit limitation rather than a demonstrated guarantee.

Table 18: AUC (%) scores for ViT-B/16 on CIFAR-10 and CIFAR-100 against various attacks ($\epsilon = 0.03$).

| Dataset | Method | FGSM | PGD | CW |
|---|---|---|---|---|
| CIFAR-10 | LID | 92.65 | 82.89 | 67.90 |
| | **Ours (RLT)** | **95.14** | **90.68** | **99.99** |
| CIFAR-100 | LID | **91.05** | 81.28 | 74.37 |
| | **Ours (RLT)** | 88.18 | **84.54** | **99.99** |

# G  Detailed Computational Cost Analysis

To substantiate our claim of low computational overhead, we provide a detailed analysis comparing our framework to two common detection paradigms: SSL-based and reference-set-based detectors. The analysis uses the target models detailed in Table 19.

First, SSL-based detectors require large, pre-trained models, which introduce a substantial and fixed overhead. As shown in Table 20, this cost is particularly prohibitive for lightweight target models, with overheads ranging from $3.21\times$ to over $200\times$ the parameters of the base classifier.

Second, reference-set detectors incur significant memory costs for storing embeddings and, in the case of graph-based methods, adjacency matrices. As detailed in Table 21, this overhead scales with the size of the reference set and can be prohibitively large, especially for graph-based approaches.

In contrast, our framework offers a uniquely flexible and tunable overhead, as demonstrated in Table 22. By adjusting the depth and width of the recovery modules, our method can achieve high detection performance with minimal cost. For instance, on ResNet110, our detector achieves a 99.57% AUC with a mere $0.24\times$ overhead—a performance drop of less than 0.3% compared to a much larger configuration. This analysis confirms that our approach is not only significantly more efficient than major alternative paradigms but also uniquely flexible, making it well-suited for deployments with lightweight target classifiers.

Table 19: Target models used in our experiments, along with their parameter counts and the number of feature blocks (layers) used for applying RT.

| Target Model | # Parameters | # Feature Blocks |
|---|---|---|
| MobileNet | 815,780 | 19 |
| ShuffleNet | 1,356,104 | 18 |
| ResNet110 | 1,730,714 | 56 |
| RepVGG | 7,956,164 | 23 |

Table 20: Evaluation of SSL-based approaches' minimum overhead ($\times$) introduced by pre-trained SSL models relative to each target model.

| SSL Models | # Params | MobileNet | ShuffleNet | ResNet110 | RepVGG |
|---|---|---|---|---|---|
| BYOL | 25,557,032 | 31.33 | 18.85 | 14.77 | 3.21 |
| SimSiam | 38,201,408 | 46.83 | 28.17 | 22.07 | 4.80 |
| MoCo v3 (ViT) | 215,678,464 | 264.38 | 159.04 | 124.62 | 27.11 |

## H   Detailed Plug-in-play System-Level Analysis

In this section, we provide a detailed theoretical and empirical analysis of the system-level performance when our detector is applied as a plug-in module to a standard classifier. We formally define the evaluation metrics, introduce a framework for establishing lower bounds on system accuracy under these metric definitions, and demonstrate how this framework can be used to select detection thresholds in practical scenarios.

### H.1   Definitions of Metrics

To formally analyze performance, we define metrics for both the base classifier and the combined classifier-detector system. Let $C(x)$ be the classifier's prediction for an input $x$, $y$ be its true label, and $D(x)$ be our detector's output, where $D(x) = 1$ signifies an adversarial detection.

- **Classifier Clean Accuracy ($CA_{cls}$)**: The accuracy of the base classifier on benign samples without any detector. $CA_{cls} = \mathbb{P}(C(x_{\text{clean}}) = y)$.

- **Classifier Robust Accuracy ($RA_{cls}$)**: The accuracy of the base classifier on adversarial samples without any detector. $RA_{cls} = \mathbb{P}(C(x_{\text{adv}}) = y)$.

- **System Clean Accuracy ($CA_{sys}$)**: The accuracy of the combined system on benign samples. A benign sample is handled correctly only if it is both correctly classified and not flagged by the detector.
$$CA_{sys} = \mathbb{E}[\mathbb{I}(C(x_{\text{clean}}) = y \wedge D(x_{\text{clean}}) = 0)]$$

- **System Robust Accuracy ($RA_{sys}$)**: The accuracy of the combined system on adversarial samples. An adversarial sample is successfully defended if it is either detected or correctly classified despite the attack.
$$RA_{sys} = \mathbb{E}[\mathbb{I}(C(x_{\text{adv}}) = y \vee D(x_{\text{adv}}) = 1)]$$

- **Overall System Accuracy ($A_{sys}$)**: The expected accuracy of the system given a probability $p$ that an input is adversarial.
$$A_{sys} = (1 - p) \cdot CA_{sys} + p \cdot RA_{sys}$$

- **False Positive Rate (FPR)**: The fraction of benign samples incorrectly flagged as adversarial. $FPR = \mathbb{P}(D(x_{\text{clean}}) = 1)$.

Table 21: Minimum overhead ($\times$) from detection approaches using a reference set with a 1024-dimensional embedding space. "Graph Structure" indicates whether the approach constructs an adjacency matrix.

| Graph Structure | Reference Set Size | # Params | MobileNet | ShuffleNet | ResNet110 | RepVGG |
|---|---|---|---|---|---|---|
| No | 1,000 | 1,024,000+ | 1.26+ | 0.76+ | 0.59+ | 0.13+ |
| | 5,000 | 5,120,000+ | 6.28+ | 3.78+ | 2.96+ | 0.64+ |
| | 40,000 | 40,960,000+ | 50.21+ | 30.20+ | 23.67+ | 5.15+ |
| Yes | 1,000 | 2,024,000+ | 2.48+ | 1.49+ | 1.17+ | 0.25+ |
| | 5,000 | 30,120,000+ | 36.92+ | 22.21+ | 17.40+ | 3.79+ |
| | 40,000 | 1,640,960,000+ | 2011.52+ | 1210.05+ | 948.14+ | 206.25+ |

Table 22: Overhead ($\times$) and AUC performance (%) (CIFAR-100 under $\ell_\infty$-PGD attack with $\epsilon = 0.02$) of our approach across different target models and varying depth and width of the recovery modules. "Performance Loss" indicates the AUC drop relative to the largest detector configuration for the same target model.

| Target Model | Depth | Width | Detector's # Params | Overhead | AUC | Performance Loss |
|---|---|---|---|---|---|---|
| MobileNet | 2 | 64 | 1,540,744 | 1.89 | 97.37 | 0.0000 |
| | 2 | 32 | 776,904 | 0.95 | 97.20 | -0.1665 |
| | 2 | 16 | 394,984 | 0.48 | 96.07 | -1.3025 |
| | 2 | 8 | 204,024 | 0.25 | 95.12 | -2.2492 |
| | 2 | 4 | 108,544 | 0.13 | 89.27 | -8.0933 |
| | 2 | 2 | 60,804 | 0.07 | 87.66 | -9.7059 |
| ShuffleNet | 2 | 128 | 2,788,840 | 2.06 | 96.07 | 0.0000 |
| | 4 | 64 | 1,548,456 | 1.14 | 95.46 | -0.6154 |
| | 2 | 64 | 1,402,664 | 1.03 | 95.96 | -0.1152 |
| | 4 | 32 | 747,656 | 0.55 | 95.12 | -0.9522 |
| | 2 | 32 | 709,576 | 0.52 | 95.48 | -0.5946 |
| | 4 | 16 | 373,368 | 0.28 | 93.88 | -2.1923 |
| | 2 | 16 | 363,032 | 0.27 | 95.25 | -0.8218 |
| | 4 | 8 | 192,752 | 0.14 | 93.02 | -3.0472 |
| | 2 | 8 | 189,760 | 0.14 | 94.56 | -1.5079 |
| ResNet110 | 3 | 256 | 5,160,992 | 2.98 | 99.84 | 0.0000 |
| | 2 | 256 | 1,514,272 | 0.87 | 99.58 | -0.2640 |
| | 3 | 64 | 652,448 | 0.38 | 99.71 | -0.1259 |
| | 2 | 64 | 416,608 | 0.24 | 99.57 | -0.2717 |
| | 3 | 16 | 158,912 | 0.09 | 99.51 | -0.3313 |
| | 2 | 16 | 142,192 | 0.08 | 99.41 | -0.4252 |
| | 3 | 4 | 75,128 | 0.04 | 99.62 | -0.2180 |
| | 2 | 4 | 73,588 | 0.04 | 99.63 | -0.2119 |
| RepVGG | 4 | 256 | 11,310,992 | 1.42 | 98.45 | 0.0000 |
| | 3 | 256 | 9,852,304 | 1.24 | 98.47 | 0.0144 |
| | 4 | 128 | 4,942,992 | 0.62 | 98.38 | -0.0721 |
| | 3 | 128 | 4,574,096 | 0.57 | 98.41 | -0.0386 |
| | 4 | 64 | 2,299,664 | 0.29 | 98.34 | -0.1092 |
| | 3 | 64 | 2,205,328 | 0.28 | 98.36 | -0.0969 |
| | 4 | 32 | 1,113,168 | 0.14 | 98.23 | -0.2185 |
| | 3 | 32 | 1,088,528 | 0.14 | 98.32 | -0.1341 |
| | 4 | 16 | 553,712 | 0.07 | 98.01 | -0.4442 |
| | 3 | 16 | 547,024 | 0.07 | 98.11 | -0.3468 |

- **True Positive Rate (TPR)**: The fraction of adversarial samples correctly flagged as adversarial. $TPR = \mathbb{P}(D(x_{\text{adv}}) = 1)$.

## H.2 Performance Analysis Across Operating Points

While AUC provides an aggregate measure, evaluating performance at fixed operating points is critical for understanding the practical trade-off between clean accuracy and robustness.

First, we establish the baseline performance of the target ResNet-110 classifier *without* our defense in Table 23. The results show that while the model achieves high clean accuracy, its robustness is completely compromised by strong attacks like PGD and AutoAttack, with the $RA_{cls}$ dropping to 0.00%.

Table 23: Baseline CA and RA (%) of the undefended ResNet-110 on CIFAR-10 ($\epsilon = 8/255$).

| Attack | $CA_{cls}$ | $RA_{cls}$ |
|---|---|---|
| FGSM | 92.49 | 25.75 |
| PGD | 92.49 | 0.00 |
| CW | 92.49 | 47.17 |
| AutoAttack | 92.49 | 0.00 |

In contrast, with our RLT detector active, the system's performance is drastically improved, as detailed in Table 24. At a modest 5% FPR, the system maintains a high $CA_{sys}$ of 88.70% while restoring the $RA_{sys}$ against PGD from 0% to 99.27%. The table further illustrates the clear trade-off available to a practitioner: increasing the FPR boosts the TPR and, consequently, the $RA_{sys}$, at the cost of $CA_{sys}$.

Table 24: Measured TPR, System Clean Accuracy ($CA_{sys}$), and System Robust Accuracy ($RA_{sys}$) (%) at varying FPRs for the defended system on CIFAR-10 ($\epsilon = 8/255$).

| Attack | Metric | @FPR5% | @FPR10% | @FPR25% | @FPR50% |
|---|---|---|---|---|---|
| | TPR | 84.84 | 94.13 | 99.21 | **99.94** |
| FGSM | $CA_{sys}$ | **88.71** | 84.52 | 71.60 | 48.80 |
| | $RA_{sys}$ | 90.80 | 96.57 | 99.58 | **99.96** |
| | TPR | 99.27 | 99.68 | 99.92 | **99.95** |
| PGD | $CA_{sys}$ | **88.70** | 84.73 | 71.71 | 48.70 |
| | $RA_{sys}$ | 99.27 | 99.68 | 99.92 | **99.95** |
| | TPR | 90.46 | 96.48 | 99.48 | **99.97** |
| CW | $CA_{sys}$ | **88.63** | 84.57 | 71.79 | 48.77 |
| | $RA_{sys}$ | 96.14 | 98.79 | 99.91 | **100.00** |
| | TPR | 88.86 | 89.90 | 91.88 | **93.87** |
| AutoAttack | $CA_{sys}$ | **88.70** | 84.73 | 71.71 | 48.70 |
| | $RA_{sys}$ | 88.86 | 89.90 | 91.88 | **93.87** |

## H.3 Plug-in Robustness Gains with Adversarially Trained Classifiers (ATC)

A key advantage of our detection framework is its role as a modular, plug-in defense. Rather than replacing robust training methods, our detector can be integrated with existing robust models, such as Adversarially Trained Classifiers (ATCs), to further enhance their performance. The value of a detector in this context is measured by the **robustness improvement** it provides to the overall system.

### H.3.1 Theoretical Justification for Robustness Improvement

The performance of a combined classifier-detector system can be formally analyzed. The expected end-to-end system robust accuracy, $RA_{sys}$, is a function of the detector's TPR and the base classifier's own robust accuracy ($RA_{cls}$). The expected system robustness is given by:

$$RA_{sys} \approx TPR + (1 - TPR) \times RA_{cls} \tag{8}$$

This can be rewritten as:

$$RA_{sys} \approx RA_{cls} + TPR \times (1 - RA_{cls}) \tag{9}$$

This relationship mathematically demonstrates that any detector with a non-zero TPR ($TPR > 0$) is expected to improve the system's robustness over the base classifier alone ($RA_{sys} > RA_{cls}$).

### H.3.2 Empirical Validation

To empirically validate this theoretical relationship, we applied our detector to several ATCs with varying levels of baseline robustness. The results, presented in Table 25, confirm two key points:

- The measured $RA_{sys}$ shows a substantial improvement over the initial $RA_{cls}$ in all cases. For instance, a classifier with a baseline robustness of 55.88% achieves a final robustness of 83.08% when paired with our detector.

- The measured $RA_{sys}$ values closely correspond to the expected values predicted by our formula, validating its utility as a model for system performance and confirming the plug-in value of our detector.

Table 25: Expected vs. Measured System Robust Accuracy ($RA_{sys}$) (%) when applying our detector to Adversarially Trained Classifiers with varying baseline robustness ($RA_{cls}$) under an adaptive attack.

| $RA_{cls}$ (%) | Detector TPR (%) | Expected $RA_{sys}$ (%) | Measured $RA_{sys}$ (%) | Improvement $\Delta$ (%) |
|---|---|---|---|---|
| 6.84 | 72.13 | 74.03 | 77.44 | +70.6 |
| 12.71 | 63.74 | 68.34 | 74.57 | +61.86 |
| 55.88 | 73.03 | 88.10 | 83.08 | +27.2 |

### H.4 Formal System Accuracy Guarantees with Lower Bounds

**General Lower Bound.** When the attack probability $p$ is unknown, we can establish a general lower bound on system accuracy.

**Theorem 5.** *Let $p, CA_{sys}, RA_{sys} \in [0, 1]$. Then $(1-p)CA_{sys} + p \cdot RA_{sys} \geq CA_{sys} \cdot RA_{sys}$.*

*Proof.* Let $f(p) = (1-p)CA_{sys} + p \cdot RA_{sys} - CA_{sys} \cdot RA_{sys}$. As a linear function of $p$ over the interval $[0, 1]$, its minimum must occur at an endpoint. At $p = 0$, $f(0) = CA_{sys}(1 - RA_{sys}) \geq 0$. At $p = 1$, $f(1) = RA_{sys}(1 - CA_{sys}) \geq 0$. Since the function is non-negative at both endpoints, the inequality holds for all $p \in [0, 1]$. $\square$

**Adaptive Lower Bound.** For practical scenarios where we can assume an upper bound on the attack probability ($p \leq p'$), we can derive a tighter lower bound.

**Theorem 6.** *Let $p \in [0, p']$ and $CA_{sys}, RA_{sys}, p' \in [0, 1]$. Then the system accuracy is lower-bounded by:*

$$(1-p)CA_{sys} + p \cdot RA_{sys} \geq CA_{sys} \cdot RA_{sys} + \max(0, (1-p')(CA_{sys} - RA_{sys}))$$

*Proof.* From Theorem 1, for any $t \in [0, p']$, we have $(1-t)CA_{sys} + t \cdot RA_{sys} \geq CA_{sys} \cdot RA_{sys}$. Let $t = p' - p$. Since $p \in [0, p']$, $t$ is also in $[0, p']$. Substituting $t = p' - p$ yields:

$$(1-p)CA_{sys} + p \cdot RA_{sys} \geq CA_{sys} \cdot RA_{sys} + (1-p')(CA_{sys} - RA_{sys})$$

This bound is tighter than the general one only when $CA_{sys} > RA_{sys}$. We therefore take the maximum of the additional term and zero to ensure the tightest possible bound in all cases.

$$(1-p)CA_{sys} + p \cdot RA_{sys} \geq CA_{sys} \cdot RA_{sys} + \max(0, (1-p')(CA_{sys} - RA_{sys}))$$

$\square$

### H.5 Empirical Validation and Optimal Threshold Selection

This framework provides a principled method for selecting an operating point under an assumed attack probability range. We applied this analysis to our FGSM attack results in Table 26. Maximizing the general lower bound ($CA_{sys} \times RA_{sys}$) suggests an FPR of 10% is optimal. However, by maximizing the Adaptive Lower Bound for a realistic low attack probability (e.g., $p' \leq 1\%$), we find that a 1% FPR provides a better lower-bound value. This confirms that our framework allows for principled configuration of the defense under practical deployment assumptions.

Table 26: Analysis of system performance and lower bounds (%) at different FPRs for the FGSM attack ($\epsilon = 8/255$).

| FPR | TPR | $CA_{sys}$ | $RA_{sys}$ | Lower Bound | Est. Lower Bound | $A_{sys}$ for attack prob. $p$ | | | | | | | | Adaptive Lower Bound for max prob. $p'$ | | | | | | | |
|---|---|---|---|---|---|---|---|---|---|---|---|---|---|---|---|---|---|---|---|---|---|
| | | | | | | 0.1% | 1% | 5% | 10% | 50% | 90% | 95% | 99% | 0.1% | 1% | 5% | 10% | 50% | 90% | 95% | 99% |
| 1 | 57.67 | **91.77** | 71.43 | 67.05 | 65.55 | **91.75** | **91.57** | **90.75** | **89.74** | 81.60 | 73.46 | 72.45 | 71.63 | **85.87** | **85.69** | **84.87** | **83.86** | 75.72 | 67.59 | 66.57 | 65.75 |
| 5 | 86.79 | 88.53 | 91.54 | 81.59 | 81.04 | 88.53 | 88.56 | 88.68 | 88.83 | 90.04 | 91.24 | 91.39 | 91.51 | 81.04 | 81.04 | 81.04 | 81.04 | 81.04 | 81.04 | 81.04 | 81.04 |
| 10 | 95.42 | 84.60 | 97.26 | **82.53** | 82.28 | 84.61 | 84.73 | 85.23 | 85.87 | **90.93** | 95.99 | 96.63 | 97.13 | 82.28 | 82.28 | 82.28 | 82.28 | **82.28** | **82.28** | **82.28** | **82.28** |
| 15 | 97.75 | 80.34 | 98.64 | 79.41 | 79.25 | 80.36 | 80.52 | 81.25 | 82.17 | 89.49 | 96.81 | 97.73 | 98.46 | 79.25 | 79.25 | 79.25 | 79.25 | 79.25 | 79.25 | 79.25 | 79.25 |
| 20 | 99.00 | 76.07 | 99.42 | 75.64 | 75.63 | 76.09 | 76.30 | 77.24 | 78.40 | 87.74 | **97.08** | **98.25** | 99.19 | 75.63 | 75.63 | 75.63 | 75.63 | 75.63 | 75.63 | 75.63 | 75.63 |
| 25 | 99.34 | 71.66 | 99.63 | 71.37 | 71.39 | 71.69 | 71.94 | 73.06 | 74.46 | 85.64 | 96.83 | 98.23 | 99.35 | 71.39 | 71.39 | 71.39 | 71.39 | 71.39 | 71.39 | 71.39 | 71.39 |
| 30 | **99.63** | 67.21 | **99.80** | 67.08 | 67.08 | 67.24 | 67.54 | 68.84 | 70.47 | 83.51 | 96.54 | 98.17 | **99.47** | 67.08 | 67.08 | 67.08 | 67.08 | 67.08 | 67.08 | 67.08 | 67.08 |

## I Robustness to Benign Noise

To assess our detector's specificity and ensure it is not merely flagging any large perturbation, we evaluated its response to significant yet benign noise. We conducted an experiment by applying random noise of varying magnitudes ($\epsilon$) to the unseen test set. Crucially, to isolate the effect of the perturbation itself from a label change, we only kept noise instances that were "benign" in their outcome, meaning they did not alter the classifier's original prediction. We then measured the new False Positive Rate (FPR) using thresholds that were originally calibrated on clean, unperturbed data.

The findings, presented in Table 27, provide strong evidence that our detector is not simply flagging any large perturbation but is specifically sensitive to the structure of adversarial attacks. At low-to-moderate noise levels ($\epsilon = 4/255$ and $8/255$), the detector remained highly stable, with the FPR remaining nearly unchanged. Even with substantial random noise ($\epsilon = 32/255$), the detector's response was moderate; for example, a threshold calibrated for 5% FPR on clean data resulted in a new FPR of only 17.51%. This suggests that the large latent shifts our method identifies are a characteristic feature of crafted, adversarial perturbations.

Table 27: Measured False Positive Rate (FPR) (%) on Benign Data with Random Noise. This table shows the new FPR when applying thresholds that were originally calibrated to give 1%, 5%, 10%, etc., FPR on clean data. The test is repeated for different magnitudes ($\epsilon$) of benign random noise.

| $\epsilon$ | FPR@1% | FPR@5% | FPR@10% | FPR@15% | FPR@20% | FPR@25% | FPR@30% | FPR@35% | FPR@40% | FPR@45% | FPR@50% |
|---|---|---|---|---|---|---|---|---|---|---|---|
| **4/255** | 0.93 | 4.77 | 9.27 | 13.60 | 18.51 | 23.03 | 27.61 | 32.01 | 36.27 | 41.20 | 46.26 |
| **8/255** | 1.05 | 4.46 | 8.04 | 11.79 | 15.65 | 19.60 | 23.52 | 27.65 | 32.07 | 36.61 | 41.15 |
| **32/255** | 12.38 | 17.51 | 22.16 | 26.71 | 31.39 | 35.99 | 40.34 | 44.72 | 49.16 | 53.54 | 57.85 |

## J Calibration-Set Sensitivity and Failure-Case Analysis

Because RLT uses empirical calibration of benign RT/LT score distributions, we additionally evaluate how sensitive the detector is to the size and coverage of the benign calibration set. Using the stored per-sample RT/LT scores from Appendix L, we recompute the empirical CDF normalization and a nominal 5% FPR threshold from randomly sampled benign calibration subsets for each dataset/model pair. We then evaluate the resulting threshold on disjoint benign samples and on FGSM, PGD, CW, and AutoAttack samples. This experiment is intended to test calibration stability; the main AUC and adaptive-attack results remain the primary detection-performance benchmarks.

Table 28 shows that the empirical calibration is stable once several hundred representative benign samples are available: with 500–1,000 samples, the held-out FPR is already close to the nominal 5% target, and increasing

Table 28: Calibration-set size sensitivity for empirical RT/LT normalization. Results are averaged over 12 dataset/model combinations and 20 random calibration resamples. The threshold is chosen for nominal 5% FPR on the sampled benign calibration set and evaluated on held-out benign samples plus adversarial samples.

| Benign calibration samples | Held-out FPR (%) | Attack detection rate (%) |
|---|---|---|
| 100 | $6.13_{\pm 2.63}$ | $83.92_{\pm 13.65}$ |
| 250 | $5.50_{\pm 1.61}$ | $87.44_{\pm 10.02}$ |
| 500 | $5.03_{\pm 1.20}$ | $87.20_{\pm 9.48}$ |
| 1,000 | $4.92_{\pm 0.87}$ | $87.77_{\pm 8.96}$ |
| 2,500 | $4.96_{\pm 0.68}$ | $88.50_{\pm 8.55}$ |
| 5,000 | $4.93_{\pm 0.59}$ | $88.48_{\pm 8.52}$ |
| 10,000 | $4.95_{\pm 0.53}$ | $88.69_{\pm 8.52}$ |

the calibration set mainly reduces variance. We also tested deliberately non-representative calibration, where the benign calibration set contains only one transformation type. As shown in Table 29, calibrating on a single benign shift can over-flag other benign shifts. Pooling representative benign variation restores near-nominal FPR for random crop, horizontal flip, and noise simultaneously. Thus, the detector should be calibrated on benign data representative of the intended deployment distribution, and recalibrated whenever the target classifier, dataset, preprocessing, or deployment distribution changes.

Table 29: Effect of benign calibration coverage. Each row uses 5,000 calibration samples and a nominal 5% FPR threshold. Single-transformation calibration intentionally under-covers benign variation and over-flags other benign shifts; pooled benign calibration restores near-nominal FPR across the benign perturbations.

| Calibration pool | FPR on crop (%) | FPR on hflip (%) | FPR on noise (%) | Attack detection rate (%) |
|---|---|---|---|---|
| Crop only | 5.00 | 26.61 | 22.31 | 96.12 |
| Horizontal flip only | 23.68 | 5.00 | 23.27 | 97.76 |
| Noise only | 65.60 | 54.25 | 5.01 | 79.83 |
| Pooled benign shifts | 5.61 | 4.26 | 4.80 | 88.41 |

We also quantify the RT/LT failure modes requested by reviewers. In Table 30, "RT-only" means RT detects the adversarial sample while LT does not; "LT rescue" means RT misses but LT detects; "RT+LT" means both detect; and "missed" means neither score crosses its benign calibrated bound. Across 12 dataset/model combinations and four attacks, RT alone catches most adversarial examples, while LT is most useful on CW and AutoAttack, where it rescues 9.56% and 11.37% of samples that RT misses. The remaining missed cases are concentrated in attacks with lower measured layer-wise violation and are the principal failure cases for future improvement.

Table 30: RT/LT failure-mode categories for adversarial samples, averaged over 12 dataset/model combinations.

| Attack | RT-only (%) | LT rescue (%) | RT+LT (%) | Missed by RT/LT (%) |
|---|---|---|---|---|
| FGSM | 93.91 | 0.01 | 0.00 | 6.08 |
| PGD | 99.94 | 0.00 | 0.00 | 0.06 |
| CW | 75.99 | 9.56 | 0.44 | 14.01 |
| AutoAttack | 79.75 | 11.37 | 0.00 | 8.88 |
| Average | 87.40 | 5.23 | 0.11 | 7.26 |

# K  Attack Configuration Summary

To make the adaptive and non-adaptive evaluations fully reproducible, we summarize the attack configurations used in the main experiments and appendices in Table 31. For the standard comparison tables, we follow the perturbation budgets used by the corresponding baselines: FGSM uses $\epsilon = 0.05$, PGD uses $\epsilon = 0.02$, and the remaining standard attacks use $\epsilon = 8/255$ unless otherwise specified. For iterative attacks,

Table 31: **Attack configurations used in the standard and adaptive evaluations.** "Default" denotes the public/default implementation recommended by the original attack or by the baseline protocol used for comparison.

| Attack | Threat model | Objective / loss | Budget | Steps / queries | Step size / search | Initialization / restarts |
|---|---|---|---|---|---|---|
| FGSM | Standard white-box | Cross-entropy untargeted | $\epsilon = 0.05$ in baseline comparison; $8/255$ in expanded validation | 1 | Single sign step | Deterministic from clean input |
| PGD | Standard white-box | Cross-entropy untargeted | $\epsilon = 0.02$ in baseline comparison; $8/255$ in expanded validation | 50 | 0.002 absolute step size | Random start; conventional/default restarts |
| CW | Standard white-box | Carlini–Wagner margin loss | $8/255$ | 50 | Default/conventional binary-search or $c$ search when available | Default initialization/restarts |
| AutoAttack | Standard ensemble | APGD-CE, APGD-DLR, FAB, Square ensemble | $8/255$ | Default standard version | Default internal schedule | Default deterministic ensemble |
| Square | Query-based black-box | Score-based square perturbation search | $8/255$ | 5,000 queries | $p_{\text{init}} = 0.8$ | 1 restart |
| Orthogonal-PGD | Adaptive white-box detector attack | Orthogonalized classification and detector-evasion directions | 0.01 and $8/255$ | Baseline protocol of Bryniarski et al. (2021); Zhiyuan et al. (2024) | Orthogonal projection removes scalar-loss balancing sensitivity | Protocol-matched initialization/restarts |
| End-to-end PGD + BPDA | Adaptive white-box detector attack | $\min_{\|\delta\|_\infty \le \epsilon}\{-\mathcal{L}_{cls}(x+\delta, y) + \lambda RLT(x+\delta)\}$ | $8/255$ | 50 | 0.002 absolute step size | Random start; 5 repeated runs; $\lambda \in \{1.00, 0.50, 0.25\}$ |
| End-to-end PGD + BPDA + ATC | Adaptive white-box detector attack with ATC | Same as above with adversarially trained classifier | $8/255$ | 50 | 0.002 absolute step size | Random start; 5 repeated runs; $\lambda \in \{1.00, 0.50, 0.25\}$ |
| SimBA | Query-based black-box adaptive check | Gradient-free logit/probability query attack | $8/255$ | 1,000 queries | Coordinate-query updates | Default query initialization |
| $L_2$-PGD | Norm-transfer appendix | Cross-entropy untargeted | $\epsilon = 0.6$ after budget search | 50 | Conventional/default $L_2$ PGD schedule | Random start |

we use 50 steps with absolute step size 0.002 when the attack implementation does not determine the step size internally. Adaptive attacks are evaluated up to $\epsilon = 8/255$.

## L  Additional Empirical Validation of the A Few Large Shifts Assumption

We add a broader empirical validation of the **A Few Large Shifts** assumption using CIFAR-10, CIFAR-100, and ImageNet with ResNet-18, ResNet-34, VGG-11, and VGG-13. For each benign or adversarial perturbation, we compute the adjacent-transition amplification $L_k(x, x') = \|z_{k+1}(x') - z_{k+1}(x)\|_2^2 / (\|z_k(x') - z_k(x)\|_2^2 + \varepsilon)$ and mark a violation when $L_k$ exceeds the benign calibrated bound for transition $k$.

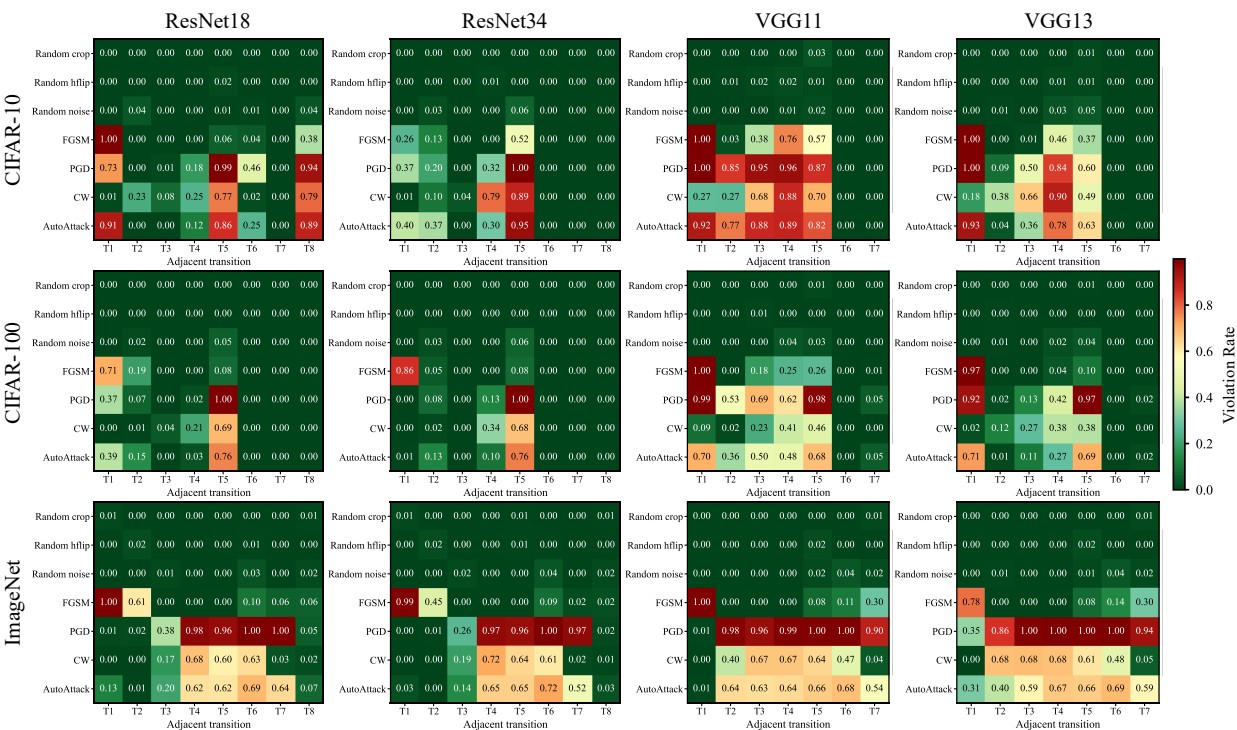

Figure 6: **Expanded layer-wise Lipschitz violation validation.** Each panel corresponds to one dataset/model pair and reports violation rates for benign perturbations (Random crop, Random hflip, Random noise) and attacks (FGSM, PGD, CW, AutoAttack). Green cells indicate low violation rates and red cells indicate high violation rates. Across all 12 panels, benign perturbations remain close to zero, while adversarial inputs trigger high violation rates at a small number of adjacent transitions, supporting the localized-shift structure of the A Few Large Shifts assumption.

## M    Detailed Algorithm

Algorithm 1 presents the detailed pseudocode for RT, LT, and the combined RLT measure.

---

**Algorithm 1** Layer-wise Adversarial Detection Measures via RT, LT, and RLT

---

**Require:** $f = f_{logit} \circ f_L \circ \cdots \circ f_1$: Target network

1: $\{R^{(L \to k)}\}_{k=k_{RT}}^{L-1}$: Trained inverse regressors

2: $\{W^{(g)}\}_{g=1}^{G}$: Learned augmentation matrices

3: $\hat{\mathcal{F}}_{RT}, \hat{\mathcal{F}}_{LT}$: Empirical CDFs of RT and LT (from benign data)

4: $\Phi^{-1}$: Standard normal quantile function

5: $x$: Test input

6: **function** RT($x$)

7:     **for** $k = k_{RT}$ to $L - 1$ **do**

8:         $e_k \leftarrow \|z_k(x) - R^{(L \to k)}(z_L(x))\|_2^2$

9:     **end for**

10:     $\boldsymbol{e}_{RT} \leftarrow (e_{k_{RT}}, \ldots, e_{L-1})$

11:     **return** $(\log(L - k_{RT}) - \mathcal{H}(\sigma(\boldsymbol{e}_{RT}))) \cdot \log\left(\frac{1}{L - k_{RT}} \sum_{k=k_{RT}}^{L-1} e_k\right)$

12: **end function**

13: **function** LT($x$)

14:     **for** $g = 1$ to $G$ **do**

15:         $\Delta z^{(g)} \leftarrow \frac{1}{L - k_{LT} + 1} \sum_{i=k_{LT}}^{L} \|z_i(x) - z_i(W^{(g)}x)\|_2^2$

16:         $\hat{y} \leftarrow \arg\max_c \sigma_c(\ell(x))$

17:         $\Delta\ell^{(g)} \leftarrow \|\mathbf{o}_{\hat{y}} - \sigma(\ell(W^{(g)}x))\|_2^2$

18:         $s^{(g)} \leftarrow \log\left(\mathcal{H}(\sigma(\ell(x))) \cdot \Delta\ell^{(g)} + \varepsilon\right) - \log\left(\Delta z^{(g)} + \varepsilon\right)$

19:     **end for**

20:     **return** $\frac{1}{G} \sum_{g=1}^{G} s^{(g)}$

21: **end function**

22: **function** RLT($x$)

23:     $r \leftarrow$ RT($x$),    $l \leftarrow$ LT($x$)

24:     $r_{norm} \leftarrow \Phi^{-1}(\hat{\mathcal{F}}_{RT}(r))$

25:     $l_{norm} \leftarrow \Phi^{-1}(\hat{\mathcal{F}}_{LT}(l))$

26:     **return** $r_{norm}^2 + l_{norm}^2$

27: **end function**

---

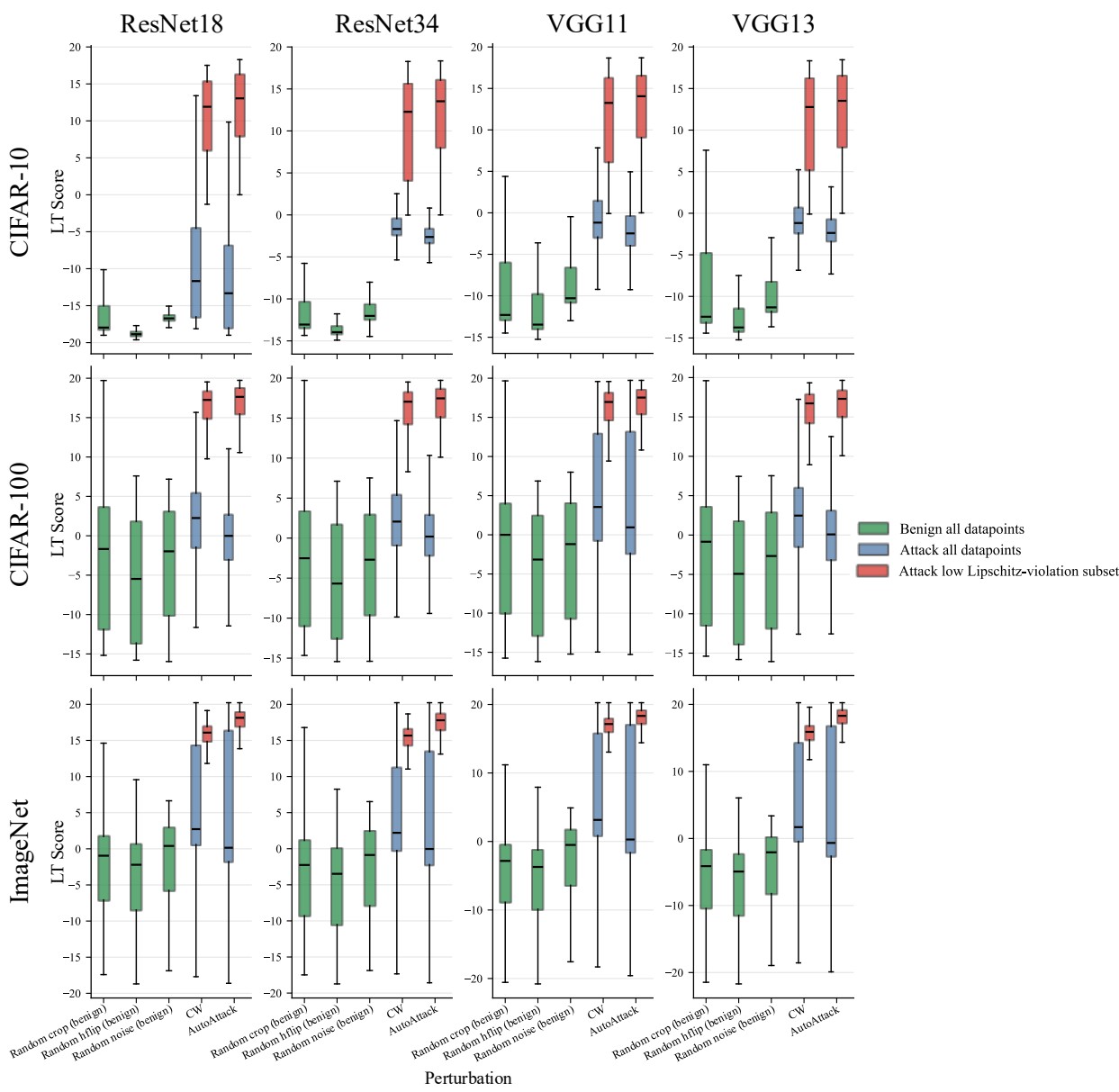

Figure 7: **LT distributions for benign perturbations and attacks with low layer-wise violation.** Each panel shows LT score distributions for benign perturbations and for CW/AutoAttack. For adversarial examples whose RT/layer-wise Lipschitz violation is low, LT scores shift upward relative to the all-sample adversarial distribution, illustrating why RT and LT are complementary and why RLT uses both signals.

Table 32: **Compact quantitative summary of expanded A Few Large Shifts validation.** "Violation" is the percentage of samples with at least one adjacent transition above the benign calibrated bound. "Peak" is the mean share of total amplification concentrated at the single largest transition. "LT rescue" is the percentage of CW/AutoAttack samples missed by the RT-style violation test but above the benign LT bound.

| Dataset | Model | Benign violation (%) | | Attack violation (%) | | Attack peak share (%) | CW/AA LT rescue (%) | Dominant transition |
|---|---|---|---|---|---|---|---|---|
| | | Mean | Max | Mean | Min | | | |
| CIFAR-10 | ResNet-18 | 3.64 | 8.82 | 97.50 | 94.40 | 89.91 | 3.68 | embedding→logit |
| | ResNet-34 | 3.27 | 8.56 | 89.78 | 69.31 | 75.84 | 3.72 | embedding→logit |
| | VGG-11 | 4.13 | 5.00 | 95.82 | 91.15 | 67.01 | 6.10 | embedding→logit |
| | VGG-13 | 4.35 | 8.73 | 96.79 | 93.25 | 62.71 | 4.63 | embedding→logit |
| CIFAR-100 | ResNet-18 | 2.36 | 6.96 | 83.46 | 72.36 | 59.80 | 0.70 | res3→res4 |
| | ResNet-34 | 3.14 | 9.00 | 84.14 | 70.92 | 50.83 | 7.90 | res3→res4 |
| | VGG-11 | 2.86 | 6.29 | 82.26 | 58.41 | 79.61 | 0.00 | vgg4→vgg5 |
| | VGG-13 | 2.74 | 7.12 | 83.94 | 63.46 | 77.57 | 0.07 | vgg4→vgg5 |
| ImageNet | ResNet-18 | 4.19 | 6.44 | 84.56 | 68.99 | 75.69 | 22.30 | res3→res4 |
| | ResNet-34 | 4.58 | 7.54 | 86.22 | 72.52 | 71.36 | 19.23 | res3→res4 |
| | VGG-11 | 3.74 | 7.55 | 84.09 | 68.16 | 53.23 | 29.03 | input→vgg1 |
| | VGG-13 | 3.86 | 7.61 | 81.58 | 69.14 | 78.44 | 28.23 | input→vgg1 |
| **Overall mean** | | **3.57** | – | **87.51** | – | **70.17** | **10.47** | – |

Table 33: **Meaning of transition labels in Figure 6.** $T_i$ denotes the $i$-th adjacent transition in the feature list recorded for each model family.

| Model family | Transition labels |
|---|---|
| CIFAR ResNet-18/34 | $T_1$: input→stem; $T_2$: stem→res1; $T_3$: res1→res2; $T_4$: res2→res3; $T_5$: res3→res4; $T_6$: res4→pool; $T_7$: pool→embedding; $T_8$: embedding→logit |
| ImageNet ResNet-18/34 | $T_1$: input→stem; $T_2$: stem→maxpool; $T_3$: maxpool→res1; $T_4$: res1→res2; $T_5$: res2→res3; $T_6$: res3→res4; $T_7$: res4→embedding; $T_8$: embedding→logit |
| CIFAR/ImageNet VGG-11/13 | $T_1$: input→vgg1; $T_2$: vgg1→vgg2; $T_3$: vgg2→vgg3; $T_4$: vgg3→vgg4; $T_5$: vgg4→vgg5; $T_6$: vgg5→embedding; $T_7$: embedding→logit |

