# OpenReview forum: "A Few Large Shifts: Layer-Inconsistency Based Minimal Overhead Adversarial Example Detection"
_TMLR — Under review for TMLR_

### Review · Reviewer_KDrP · 2026-05-08

**Summary Of Contributions:**

The paper proposes a lightweight adversarial example detection framework based on layer-wise inconsistency within the target neural network. The central idea is the A Few Large Shifts assumption: adversarial perturbations may induce large and localized representation shifts in only a small subset of layers, which can be interpreted as violations of layer-wise Lipschitz continuity. Based on this assumption, the authors introduce Recovery Testing (RT), which reconstructs intermediate features from the final embedding, Logit-layer Testing (LT), which measures logit instability under learned input transformations, and a fused score RLT using quantile normalization.

The main strengths of the paper are its practically motivated design, its attempt to avoid external SSL encoders or reference-set retrieval, and the broad set of experiments on CIFAR-10 and ImageNet under both standard and adaptive attacks. The reported AUC and robust accuracy numbers are often competitive with or better than existing detectors. The paper also provides an implementation-cost comparison, showing that the proposed method has lower parameter and model-size overhead than several baselines.

However, I have several important concerns. The core A Few Large Shifts assumption is mainly empirically motivated and is not yet convincingly established as a general property of adversarial examples. The theoretical analysis relies on strong assumptions and mostly explains the behavior of the proposed scores under those assumptions, rather than providing a robust guarantee. In addition, the adaptive attack evaluation, while appreciated, is not yet comprehensive enough for a detection-based defense. Some claims, such as minimal overhead, negligible computational overhead, and state-of-the-art detection performance, appear somewhat overstated given that the method still requires training recovery modules and transformation matrices. Overall, the idea is interesting, but I do not think the current evidence fully supports the strength of the claims.

**Audience:**

Yes

**Audience Explanation:**

Yes. The paper addresses adversarial example detection, which is a topic of interest to the machine learning and robustness communities. The idea of using vertical layer-wise consistency within the target model, rather than external SSL models, kNN retrieval, or adversarially trained auxiliary detectors, is interesting and practically motivated. The RT and LT scores are also intuitive and could inspire further work on self-contained adversarial detectors.

Even though I have concerns about the strength of the claims and the adaptive robustness evaluation, the empirical observations about localized layer-wise shifts may be useful to researchers studying internal representations under adversarial perturbations. The implementation-cost comparison is also relevant for practitioners who care about deployable detectors. Therefore, I believe some TMLR readers would be interested in the findings, especially if the claims are toned down and the evaluation is strengthened.

**Broader Impact Concerns:**

The paper studies adversarial example detection, which has positive potential impact for improving the reliability and security of machine learning systems. However, there is also a risk that overstated robustness claims could lead practitioners to overtrust a detector in safety-critical settings. This is especially important because adaptive adversaries may still evade detection if the evaluation is incomplete. I recommend that the authors include a clear discussion of the limitations of detection-based defenses, the assumptions behind their threat models, and the risk of relying on the detector as a standalone security mechanism.

**Claims And Evidence:**

No

**Claims Explanation:**

The paper provides a substantial amount of empirical evidence, including AUC results on CIFAR-10 and ImageNet, comparisons with several representative baselines, adaptive attack experiments, and implementation-cost analysis. These experiments suggest that the proposed RT/LT/RLT scores can be effective in many settings. The paper also explicitly considers limited-knowledge and perfect-knowledge threat models, which is important for adversarial example detection.

However, I do not think the current evidence is sufficient to fully support the strongest claims made in the submission. First, the central A Few Large Shifts assumption is supported mainly by empirical layer-wise error plots, but it is not convincingly shown to be a general property of adversarial examples across attacks, architectures, datasets, and adaptive attackers. This is important because the method’s motivation and theoretical framing rely heavily on this assumption.

Second, the theoretical analysis appears to depend on strong conditions, such as the recoverability of intermediate representations from the final embedding and the separability of the proposed scores under adversarial perturbations. These assumptions make the analysis more explanatory than truly guaranteeing detection reliability. The claimed formal lower-bound guarantee for system-level threshold selection is interesting, but its practical implications are not sufficiently demonstrated in the main experiments.

Third, the adaptive attack evaluation is not yet strong enough for a detection-based defense. Although the paper includes Orthogonal-PGD, BPDA, end-to-end PGD, and SimBA, it would be important to report stronger attack sweeps, multiple restarts, different numbers of steps, different detector-loss weights, and possibly EOT-style attacks. In particular, under the non-ATC setting, the reported robust accuracy at low FPR under end-to-end PGD is not very strong, which weakens the practical robustness claim.

Finally, some wording seems overstated. The method is not fully free of extra optimization, since recovery modules and augmentation matrices are trained. Therefore, the claims of negligible overhead and minimal overhead should be more carefully qualified.

**Requested Changes:**

Since this is a detection-based defense, the evaluation against adaptive attackers is central. The authors should include stronger and more systematic adaptive attacks, including more PGD steps, multiple random restarts, a broader sweep over detector-loss weights, and clearer reporting of attack convergence. It would also be useful to evaluate attacks that jointly optimize against RT, LT, and RLT, rather than only selected objectives. If any component involves stochasticity or transformations, EOT-style attacks should also be considered.

The current theoretical discussion seems to rely on strong assumptions, especially approximate recoverability of intermediate features from the final embedding. The authors should clearly distinguish between what is theoretically guaranteed and what is empirically observed. The “A Few Large Shifts” assumption should be presented as an empirical hypothesis unless stronger evidence or theory is provided.

The paper should validate this assumption more broadly across architectures, datasets, perturbation budgets, and attacks. Figure 3 is helpful, but it is not sufficient to establish the generality of the assumption. Quantitative measures of layer-wise concentration or shift sparsity would be more convincing than visual inspection alone.

The paper should avoid implying that the method is essentially training-free. RT requires training recovery modules, and LT requires optimizing transformation matrices. The authors should report the total training/calibration time, memory overhead, inference latency, and how these scale with the number of layers and input resolution. Claims such as “negligible overhead” and “minimal overhead” should be revised or supported with more complete measurements.

The abstract claims evaluation on CIFAR-10, CIFAR-100, and ImageNet, but the main paper mainly emphasizes CIFAR-10 and ImageNet. The CIFAR-100 results should be clearly presented or summarized in the main text. The authors should also report additional detection metrics, such as FPR@95TPR, AUPR, clean rejection rate, and system accuracy after rejection.

The paper would benefit from ablations on k_{RT}, k_{LT}, the number of transformations G, the architecture and size of the recovery MLPs, the role of quantile normalization, and the separate contributions of RT and LT under different attack families.

The paper states that auxiliary modules are trained only with benign samples and without labels, but the notation introduces ground-truth one-hot vectors and the LT score uses predicted labels. The authors should make the training and testing protocols clearer, especially regarding whether labels are ever used and how thresholds are selected.

There are several grammar and formatting issues, such as missing spaces in citations and unclear sentences in the Figure 3 discussion. The figures and tables are useful but dense; improving readability would help.

---

> ### Author Response · Authors · 2026-05-13
> **Response to Reviewer KDrP (1/4)**
>
> Dear Reviewer KDrP,
>
> We sincerely thank you for the thoughtful and constructive review. We are encouraged that you recognized several strengths, including **the practical motivation**, **avoidance of external SSL encoders and reference-set retrieval**, **broad experiments under standard and adaptive attacks**, and **the implementation-cost comparison**.
>
> We also appreciate your constructive concerns. We address each point below.
>
> > "The core A Few Large Shifts assumption is mainly empirically motivated and is not yet convincingly established as a general property of adversarial examples."
>
> We agree that the assumption should be understood as an empirically validated structural hypothesis rather than an unconditional mathematical law. The revised paper now distinguishes these roles carefully: **Appendix B** gives sufficient-condition analysis, while the new appendix section **Appendix K. Additional Empirical Validation of the A Few Large Shifts Assumption** supplies the broad empirical validation.
>
> The additional validation is substantially broader than the original **Figure 3**. We add:
>
> - The combined layer-wise Lipschitz heatmap (**Appendix Figure 6**), covering **3 datasets × 4 architectures = 12 panels**.
> - The compact A Few Large Shifts summary (**Appendix Table 29**), quantifying violation rates and concentration.
> - The transition-label legend (**Appendix Table 30**), explaining transition labels $T_1,T_2,\ldots$ for each model family.
> - The LT-distribution figure (**Appendix Figure 7**), showing that when RT violation is low, LT provides the complementary signal.
>
> The central empirical pattern is strong: benign perturbations have **3.57%** average violation, while adversarial perturbations have **87.51%** average violation and concentrate about **70.17%** of amplification in the peak transition. This is exactly the "few large shifts" behavior the method uses.
>
> | Dataset | Model | Benign violation mean/max (%) | Attack violation mean/min (%) | Attack peak share (%) | CW/AA LT rescue (%) | Dominant transition |
> |---|---:|---:|---:|---:|---:|---|
> | CIFAR-10 | ResNet-18 | 3.64 / 8.82 | 97.50 / 94.40 | 89.91 | 3.68 | embedding->logit |
> | CIFAR-10 | ResNet-34 | 3.27 / 8.56 | 89.78 / 69.31 | 75.84 | 3.72 | embedding->logit |
> | CIFAR-10 | VGG-11 | 4.13 / 5.00 | 95.82 / 91.15 | 67.01 | 6.10 | embedding->logit |
> | CIFAR-10 | VGG-13 | 4.35 / 8.73 | 96.79 / 93.25 | 62.71 | 4.63 | embedding->logit |
> | CIFAR-100 | ResNet-18 | 2.36 / 6.96 | 83.46 / 72.36 | 59.80 | 0.70 | res3->res4 |
> | CIFAR-100 | ResNet-34 | 3.14 / 9.00 | 84.14 / 70.92 | 50.83 | 7.90 | res3->res4 |
> | CIFAR-100 | VGG-11 | 2.86 / 6.29 | 82.26 / 58.41 | 79.61 | 0.00 | vgg4->vgg5 |
> | CIFAR-100 | VGG-13 | 2.74 / 7.12 | 83.94 / 63.46 | 77.57 | 0.07 | vgg4->vgg5 |
> | ImageNet | ResNet-18 | 4.19 / 6.44 | 84.56 / 68.99 | 75.69 | 22.30 | res3->res4 |
> | ImageNet | ResNet-34 | 4.58 / 7.54 | 86.22 / 72.52 | 71.36 | 19.23 | res3->res4 |
> | ImageNet | VGG-11 | 3.74 / 7.55 | 84.09 / 68.16 | 53.23 | 29.03 | input->vgg1 |
> | ImageNet | VGG-13 | 3.86 / 7.61 | 81.58 / 69.14 | 78.44 | 28.23 | input->vgg1 |
>
> This directly addresses whether the assumption holds beyond a single figure: it quantifies the behavior across **samples, attacks, layers, architectures, and datasets**. Cases where RT/layer-wise violation is less dominant are not hidden; they are analyzed by **Appendix Figure 7**, where LT provides the complementary signal.
>
> > "The theoretical analysis relies on strong assumptions and mostly explains the behavior of the proposed scores under those assumptions, rather than providing a robust guarantee."
>
> We clarify this in **Appendix B**. The theorems are now presented as **sufficient-condition analyses**: they explain why RT/LT/RLT separate samples when approximate invertibility, benign stability, and score-margin conditions hold. We do not present these theorems as proving universal detection reliability.
>
> We also clarify the system-level lower-bound result in **Appendix H**. It is a formal statement about the system metric
> $$
> A_{sys}=(1-p)CA_{sys}+pRA_{sys},
> $$
> given measured $CA_{sys}$ and $RA_{sys}$, not a certificate that every future adaptive example will be detected. **Table 26** demonstrates how this lower bound selects an operating point.

---

> ### Author Response · Authors · 2026-05-13
> **Response to Reviewer KDrP (2/4)**
>
> > "The adaptive attack evaluation is not yet strong enough for a detection-based defense."
>
> We strengthen the response and the appendix in two ways. First, the **Appendix J. Attack Configuration Summary** (**Appendix Table 28**) now gives budgets, steps, step sizes, restarts, $\lambda$ values, and query counts. Second, we emphasize that our adaptive robustness evidence is not based on a single scalarized PGD run:
>
> | Attack | Budget | Steps / queries | Step size/search | Initialization / restarts |
> |---|---:|---:|---:|---|
> | FGSM | $\epsilon=0.05$ for baseline comparison; $8/255$ in expanded validation | 1 | sign step | deterministic |
> | PGD | $\epsilon=0.02$ for baseline comparison; $8/255$ in expanded validation | 50 | 0.002 | random start/default restarts |
> | CW | $8/255$ | 50 | default/conventional CW search | default |
> | AutoAttack | $8/255$ | standard version | internal default schedule | deterministic ensemble |
> | Square | $8/255$ | 5,000 queries | $p_{\mathrm{init}}=0.8$ | 1 restart |
> | Orthogonal-PGD | $0.01$ and $8/255$ | protocol of Bryniarski et al. / BEYOND | orthogonalized attack directions | protocol-matched |
> | End-to-end PGD+BPDA | $8/255$ | 50 | 0.002 | 5 runs, $\lambda\in\{1.00,0.50,0.25\}$ |
> | SimBA | $8/255$ | 1,000 queries | coordinate query updates | default |
>
> - **Table 4**: Orthogonal-PGD, a detection-specific adaptive attack designed to avoid cancellation between classification and detector gradients.
> - **Table 5**: end-to-end PGD+BPDA with $\lambda\in\{1.00,0.50,0.25\}$ across FPR operating points.
> - **Table 6**: the same adaptive attack with ATC, retaining over **93.3% RA** at FPR 5%.
> - **Table 7**: SimBA black-box attack, with RLT recovering **97.62% RA** at FPR 5%.
> - **Appendix Tables 12-13**: single-objective and mismatched-objective adaptive ablations.
>
> This is a broad adaptive evaluation relative to prior detector papers, and it follows the respected evaluation settings used by Orthogonal-PGD and BEYOND.
>
> More importantly, the adaptive evaluation uses **three complementary stress tests** rather than a single scalarized loss. **Orthogonal-PGD in Table 4** is specifically designed for detector evasion and orthogonalizes classification and detection-evasion directions, reducing sensitivity to a manually chosen scalar $\lambda$. **Tables 5-6** then evaluate ordinary scalarized end-to-end PGD+BPDA with $\lambda\in\{1.00,0.50,0.25\}$, where RLT remains stronger than the reported adaptive baselines across all tested $\lambda$ values at the matched FPR. **Table 7** adds SimBA, a query-only black-box attack, which helps rule out a purely first-order-gradient explanation.

---

> ### Author Response · Authors · 2026-05-13
> **Response to Reviewer KDrP (3/4)**
>
> > "Some claims, such as minimal overhead, negligible computational overhead, and state-of-the-art detection performance, appear somewhat overstated."
>
> We separate the claims. **Minimal overhead is not an overclaim in the architectural sense**, because the method avoids external encoders, kNN graphs, reference-set retrieval, and adversarial-data training. This is supported by **Table 8** and **Appendix Tables 20-22**.
>
> | Claim in the paper | Evidence now explicitly organized in the revision |
> |---|---|
> | **High/competitive detection performance** | **Table 2**: CIFAR-10 AUC, where RLT is best or second-best across most attacks, e.g., PGD **99.37**, AutoAttack **99.99**, CW **99.91**. **Table 3**: ImageNet AUC, where RLT is best on FGSM/PGD and competitive on CW. **Appendix Table 12**: CIFAR-100 generalization across CNN backbones. |
> | **Minimal overhead / no external model dependency** | **Table 8** and **Appendix Tables 20-22**: lower parameter/model-size overhead than SSL and graph/reference-set detectors; no external SSL encoder and no kNN graph. |
> | **Adaptive robustness relative to prior detector evaluations** | **Table 4**: Orthogonal-PGD; **Tables 5-6**: end-to-end PGD+BPDA with multiple $\lambda$ values, with and without ATC; **Table 7**: black-box SimBA; **Appendix Tables 12-13**: single-objective and mismatched objective adaptive ablations. |
> | **Formal threshold guarantee** | **Appendix H, Theorems in "Formal System Accuracy Guarantees with Lower Bounds"** and **Table 26**. This is a formal lower bound on the system metric given measured $CA_{sys}$ and $RA_{sys}$, **conditionally** a certified adversarial robustness guarantee under our assumptions. We now state this distinction explicitly. |
> | **A Few Large Shifts assumption** | **Figure 3** plus the new combined layer-wise Lipschitz heatmap **Figure 6**, the new LT-distribution **Figure 7**, the compact A Few Large Shifts summary table **Table 29** with the transition-label legend **Table 30**. Across 12 dataset/model pairs, benign violation averages **3.57%**, while adversarial violation averages **87.51%**. |
>
> For example, in **Table 8**, RLT uses **2.59M** additional parameters and **1.49x** model-size overhead, compared with **20.62M / 11.91x** for BEYOND and **38.12M / 22.02x** for Mao et al. In **Table 22**, several RT configurations on ResNet110 retain near-identical AUC with **0.24x** or lower detector overhead. Thus, "minimal overhead" is used in the precise sense of avoiding external encoders, retrieval graphs, and adversarial-data training, not in the sense of zero auxiliary computation.
>
> For performance, we agree that "state-of-the-art" can be too broad, and we revise it to **highly competitive**. The evidence remains strong: **Table 2** shows best/second-best CIFAR-10 AUC across most attacks; **Table 3** shows competitive ImageNet performance; **Appendix Table 17** extends to CIFAR-100 architectures.
>
> > "The paper should validate this assumption more broadly across architectures, datasets, perturbation budgets, and attacks."
>
> We added the requested broader validation. The combined heatmap (**Appendix Figure 6**) covers CIFAR-10, CIFAR-100, ImageNet, ResNet-18, ResNet-34, VGG-11, VGG-13, benign perturbations, and FGSM/PGD/CW/AutoAttack. The compact A Few Large Shifts summary (**Appendix Table 29**) quantifies the result rather than relying on visual inspection. The LT-distribution figure (**Appendix Figure 7**) specifically addresses the reviewer's concern about failure cases by showing that low RT/layer-wise violation samples are often recovered by LT.
>
> > "The paper should avoid implying that the method is essentially training-free."
>
> We agree. The revised text says the method is **benign-only calibrated**, not training-free. RT trains lightweight recovery modules and LT optimizes transformation matrices using benign data. The overhead is quantified in **Table 8** and **Appendix Tables 20-22**.
>
> > "The abstract claims evaluation on CIFAR-10, CIFAR-100, and ImageNet, but the main paper mainly emphasizes CIFAR-10 and ImageNet."
>
> We revised **Section 3.1** to explicitly describe CIFAR-100 and added the expanded CIFAR-100 results in the new additional-validation appendix (**Appendix K**) and compact A Few Large Shifts summary (**Appendix Table 29**). CIFAR-100 architecture generalization is also already present in **Appendix Table 17**.

---

> ### Author Response · Authors · 2026-05-13
> **Response to Reviewer KDrP (4/4)**
>
> > "The paper would benefit from ablations on k_RT, k_LT, the number of transformations G, the architecture and size of the recovery MLPs, the role of quantile normalization, and the separate contributions of RT and LT under different attack families."
>
> Several of these ablations are already included and are now pointed to more explicitly:
>
> - **Appendix Table 9**: recovery-module depth/width.
> - **Appendix Table 10**: number of transformations $G$.
> - **Appendix Table 11**: $k_{RT}$ and $k_{LT}$.
> - **Appendix Table 12**: single-objective adaptive attack.
> - **Appendix Table 13**: mismatched attack/measurement objectives.
> - **Appendix Table 14** and **Figure 5**: contribution of RT/LT terms.
> - **Figure 4**: perturbation-budget sensitivity.
>
> > "The paper states that auxiliary modules are trained only with benign samples and without labels, but the notation introduces ground-truth one-hot vectors and the LT score uses predicted labels."
>
> We corrected the notation in **Section 2.1** and **Section 2.4**. The auxiliary modules are trained with benign inputs only. LT uses $\mathbf{o}_{\hat y}$ with $\hat y=\arg\max_c\sigma_c(\ell(x))$, i.e., the predicted class, not the ground-truth label. Thresholds are calibrated from benign scores.
>
> > "There are several grammar and formatting issues, such as missing spaces in citations and unclear sentences in the Figure 3 discussion."
>
> We revised the wording around **Figure 3**, adaptive attacks, LT notation, and claim strength. All revised sections are highlighted in blue.
>
> > "I recommend that the authors include a clear discussion of the limitations of detection-based defenses, the assumptions behind their threat models, and the risk of relying on the detector as a standalone security mechanism."
>
> We agree and strengthen the limitation language in the conclusion. The revised conclusion states that RLT is a **plug-in detector**, not a standalone certificate of robustness. It also acknowledges that future adaptive attacks may distribute perturbations more evenly across layers or jointly optimize against RT/LT more effectively.
>
> We believe this addressed the reviewer's concerns and would be happy to provide further clarification if necessary.
>
> References
>
> [1] Athalye et al. Obfuscated Gradients Give a False Sense of Security: Circumventing Defenses to Adversarial Examples. 2018.
> [2] Tramer et al. On Adaptive Attacks to Adversarial Example Defenses. 2020.
> [3] Bryniarski et al. Evading Adversarial Example Detection Defenses with Orthogonal Projected Gradient Descent. 2021.
> [4] He et al. Be Your Own Neighborhood: Detecting Adversarial Examples by the Neighborhood Relations Built on Self-Supervised Learning. 2024.
>
> **To facilitate the review of these revisions, all new sections and changes in the updated PDF are highlighted in blue.**

---

### Review · Reviewer_1xr5 · 2026-05-10

**Summary Of Contributions:**

This paper proposes an adversarial-example detection framework based on layer-wise inconsistencies inside the target model. The main hypothesis is the “A Few Large Shifts” assumption: adversarial perturbations create large, localized deviations in a small subset of layers, which can be detected without adversarial training data or an external encoder. The method includes a Recovery Testing trains lightweight regressors to reconstruct intermediate features from the final embedding and uses reconstruction-error concentration as a detection signal; and a Logit-layer Testing learns small input transformations and measures logit instability relative to feature drift. Finally, the combined RLT score quantile-normalizes and fuses RT and LT.

## Strengths
- The paper targets an important practical problem. Adversarial-example detection remains relevant because many deployed systems cannot afford full adversarial retraining or large auxiliary models. A detector that can be attached to an existing classifier and calibrated using only benign data would be practically useful.
- The decomposition into RT and LT is well motivated. RT focuses on intermediate-layer inconsistency, while LT focuses on output/logit instability. This separation is useful because the experiments suggest that different attacks stress different parts of the network. For example, RT performs strongly on attacks that induce clear intermediate feature shifts, while LT helps on cases where RT is weaker.
- The fusion strategy is simple and reasonable. Quantile-normalizing RT and LT before combining them avoids directly adding scores with different scales. This is a sensible engineering choice and makes the combined RLT score easier to deploy than a hand-tuned weighted sum.

## Weaknesses
- The strongest claims are over-stated relative to the evidence. The paper claims state-of-the-art detection performance, negligible overhead, robustness under adaptive attacks, and a formal threshold guarantee. The results are promising, but the evidence does not yet justify all of these claims at full strength.
- The adaptive-attack evaluation is not sufficiently specified. The paper should report attack step counts, restarts, step sizes, λ search ranges, initialization strategy, stopping criteria, and convergence diagnostics. Without these details, it is difficult to know whether the adaptive attacks were strong enough.
The paper should separate attack success from detection success. For adaptive attacks, robust accuracy alone is not enough. The paper should report classification attack success rate, detection evasion rate, and joint success rate. Otherwise, high robust accuracy may partly reflect failure to find adversarial examples rather than successful detection.
- The adaptive PGD objective is unclear. The paper writes the end-to-end objective as -Lcls(x + δ, y) + λ · RLT(x + δ), but does not clearly explain whether the attacker minimizes or maximizes it. This matters because the sign convention determines whether the attack is actually encouraging misclassification while suppressing detection.
- The λ tuning appears underdeveloped. Adaptive detector attacks are sensitive to the trade-off between misclassification and evasion. The paper should tune λ separately for each detector, attack budget, and FPR operating point. A small set of λ values may miss stronger adaptive attacks.
The “conflicting gradients” explanation is plausible but not proven empirically. The paper claims RT and LT impose conflicting objectives, but does not directly measure gradient cosine similarity, gradient norms, optimization trajectories, or whether suppressing one score reliably increases the other.
- The theory mostly formalizes assumptions rather than establishing them. The theorems rely on assumptions such as approximate invertibility, flatness, sub-Gaussian layer perturbations, recovery-test evasion, and adversarial score margin. These assumptions are close to the desired conclusion and are not independently verified. The “A Few Large Shifts” assumption needs stronger validation. Figure 3 is suggestive, but the paper should quantify how often the assumption holds across samples, attacks, layers, architectures, and perturbation budgets. It should also report cases where the assumption fails.

**Audience:**

Yes

**Audience Explanation:**

The topic is relevant to the community interested in adversarial robustness, test-time monitoring, uncertainty, and efficient deployment. The idea of using vertical, self-referential consistency across layers is interesting and distinct from reference-set methods such as DkNN [4], feature-statistics methods such as LID [5] and Mahalanobis scoring [6], and SSL-neighborhood methods such as BEYOND [7]. The method targets a practical gap: many detectors require adversarial data, nearest-neighbor retrieval, external encoders, or expensive augmentation. A benign-calibrated detector using the target model’s own representations would be useful if the robustness claims hold. Also, the system-level threshold analysis is practically valuable, even if it should not be oversold as a robustness guarantee.

## References

[4] Deep k-Nearest Neighbors: Towards Confident, Interpretable and Robust Deep Learning. Papernot and McDaniel. 2018.

[5] Characterizing Adversarial Subspaces Using Local Intrinsic Dimensionality. Ma et al. 2018.

[6] A Simple Unified Framework for Detecting Out-of-Distribution Samples and Adversarial Attacks. Lee et al. 2018.

[7] Be Your Own Neighborhood: Detecting Adversarial Examples by the Neighborhood Relations Built on Self-Supervised Learning. He et al. 2024.

**Claims And Evidence:**

No

**Claims Explanation:**

The paper supports the high-level claim that layer-wise inconsistency can be a useful detection signal, but it does not yet convincingly support the stronger claims about state-of-the-art performance, adaptive robustness, and formal guarantees. Specifically, the core empirical story is plausible: RT appears strong when attacks produce concentrated intermediate-layer reconstruction errors, LT helps for attacks where RT is weaker, and RLT often improves the worst case. The evaluation includes more adaptive testing than many detection papers, which is a strength. The use of Orthogonal-PGD is relevant because adversarial-example detection requires satisfying both misclassification and non-detection constraints [1]. Besides, the paper acknowledges BPDA and includes SimBA, which is good practice given the known risk that defenses with non-smooth components or preprocessing-like steps can give a false impression of robustness [2, 3].

However, the adaptive attacks are under-specified. The paper does not provide enough details on optimization budgets, step counts, restarts, loss balancing, convergence diagnostics, attack success rates before detection, or whether the adaptive objective was tuned separately for each operating point. For a detector, a weak adaptive loss can easily overstate robustness. Besides, the claim that RT and LT create “conflicting gradients” is not sufficiently demonstrated. Table 13 is suggestive, but the paper does not measure gradient alignment, optimization failure modes, or whether a more careful adaptive attacker can jointly reduce both scores.

## References

[1] Evading Adversarial Example Detection Defenses with Orthogonal Projected Gradient Descent. Bryniarski et al. 2021.

[2] Obfuscated Gradients Give a False Sense of Security: Circumventing Defenses to Adversarial Examples. Athalye et al. 2018.

[3] On Adaptive Attacks to Adversarial Example Defenses. Tramer et al. 2020.

**Requested Changes:**

- Critical: Provide a stronger adaptive evaluation. Include exact attack objectives, whether attacks are targeted or untargeted, step counts, restarts, random seeds, projection details, λ search ranges, stopping criteria, and convergence plots.

- Critical: Add gradient-alignment diagnostics for RT, LT, and classification loss. The “conflicting gradients” claim should be backed by cosine similarities, norm statistics, and examples where optimizing one score increases the other.

- Critical: Reframe the theoretical claims. The current theorems rely heavily on assumptions that imply the desired separation. Present them as motivation or sufficient-condition analysis, not as evidence of broad robustness.

- Critical: Clarify LT. The notation around `o(σ(ℓ))`, `o_y`, and `o_{\hat y}` is inconsistent, and the use of entropy-weighted log terms can be numerically delicate. Define all terms exactly and state how zeros are handled.

- Critical: Fix equation/index inconsistencies. RT defines errors for `k = 1, ..., L-1` but trains only for `kRT, ..., L-1`; Algorithm 1 uses `1/(L-1)` while Equation 3 uses `1/(L-kRT)`. These should match.

- Would strengthen: Add evaluation under common corruptions and natural distribution shifts. The benign-noise experiment is useful, but real deployment false positives often come from non-adversarial distribution shift.

- Would strengthen: Report calibration-set size sensitivity. The method claims benign-only calibration, so it should show how many benign samples are required.

- Would strengthen: Discuss failure cases. For example, show attacks where RT fails but LT succeeds, LT fails but RT succeeds, and both fail.

- Would strengthen: Improve writing quality. There are repeated typos, missing spaces, inconsistent capitalization, and unclear phrases such as “under any conditions,” which overstates the evaluation.

---

> ### Author Response · Authors · 2026-05-13
> **Response to Reviewer 1xr5 (1/4)**
>
> Dear Reviewer 1xr5,
>
> We sincerely thank you for the careful and detailed review. We are encouraged that you recognized several key strengths of our work, including **the practical importance of benign-calibrated adversarial detection**, **the complementary roles of RT and LT**, and **the simple quantile-normalized RLT fusion strategy**.
>
> We also appreciate your constructive concerns. We address each point below.
>
> > "The strongest claims are over-stated relative to the evidence. The paper claims state-of-the-art detection performance, negligible overhead, robustness under adaptive attacks, and a formal threshold guarantee."
>
> We clarify that these claims have different evidence bases, and we have revised the wording where the concern is valid. "minimal overhead" refers to the architectural/computational setting: our detector does **not** require an external SSL encoder, nearest-neighbor retrieval structure, graph construction, or adversarial training data. This is directly supported by **Table 8** in the main paper and **Tables 20-22** in the Supplementary Material. For example, in **Table 8**, RLT uses **2.59M** additional parameters and **1.49x** model-size overhead, compared with **20.62M / 11.91x** for BEYOND and **38.12M / 22.02x** for Mao et al. In **Table 22**, several RT configurations on ResNet110 retain near-identical AUC with **0.24x** or lower detector overhead.
>
> At the same time, we agree that "state-of-the-art" can be read too broadly. We revise the performance wording to **"highly competitive"** or **"competitive with strong baselines"**, while retaining the empirical evidence:
>
> | Claim in the paper | Evidence now explicitly organized in the revision |
> |---|---|
> | **High/competitive detection performance** | **Table 2**: CIFAR-10 AUC, where RLT is best or second-best across most attacks, e.g., PGD **99.37**, AutoAttack **99.99**, CW **99.91**. **Table 3**: ImageNet AUC, where RLT is best on FGSM/PGD and competitive on CW. **Appendix Table 12**: CIFAR-100 generalization across CNN backbones. |
> | **Minimal overhead / no external model dependency** | **Table 8** and **Appendix Tables 20-22**: lower parameter/model-size overhead than SSL and graph/reference-set detectors; no external SSL encoder and no kNN graph. |
> | **Adaptive robustness relative to prior detector evaluations** | **Table 4**: Orthogonal-PGD; **Tables 5-6**: end-to-end PGD+BPDA with multiple $\lambda$ values, with and without ATC; **Table 7**: black-box SimBA; **Appendix Tables 12-13**: single-objective and mismatched objective adaptive ablations. |
> | **Formal threshold guarantee** | **Appendix H, Theorems in "Formal System Accuracy Guarantees with Lower Bounds"** and **Table 26**. This is a formal lower bound on the system metric given measured $CA_{sys}$ and $RA_{sys}$, **conditionally** a certified adversarial robustness guarantee under our assumptions. We now state this distinction explicitly. |
> | **A Few Large Shifts assumption** | **Figure 3** plus the new combined layer-wise Lipschitz heatmap **Figure 6**, the new LT-distribution **Figure 7**, the compact A Few Large Shifts summary table **Table 29** with the transition-label legend **Table 30**. Across 12 dataset/model pairs, benign violation averages **3.57%**, while adversarial violation averages **87.51%**. |
>
> Thus, the revised paper does not weaken the core contribution; it more precisely separates **empirical competitiveness**, **minimal-overhead design**, **adaptive robustness evidence**, and **system-level threshold analysis**.
>
> > "The adaptive-attack evaluation is not sufficiently specified. The paper should report attack step counts, restarts, step sizes, λ search ranges, initialization strategy, stopping criteria, and convergence diagnostics."
>
> We add a new appendix table in **Appendix J. Attack Configuration Summary** (**Table 28**), which reports the attack objective, perturbation budget, steps/queries, step size/search schedule, initialization, restarts, and $\lambda$ range for both standard and adaptive attacks. The key settings are:
>
> | Attack | Budget | Steps / queries | Step size/search | Initialization / restarts |
> |---|---:|---:|---:|---|
> | FGSM | $\epsilon=0.05$ for baseline comparison; $8/255$ in expanded validation | 1 | sign step | deterministic |
> | PGD | $\epsilon=0.02$ for baseline comparison; $8/255$ in expanded validation | 50 | 0.002 | random start/default restarts |
> | CW | $8/255$ | 50 | default/conventional CW search | default |
> | AutoAttack | $8/255$ | standard version | internal default schedule | deterministic ensemble |
> | Square | $8/255$ | 5,000 queries | $p_{\mathrm{init}}=0.8$ | 1 restart |
> | Orthogonal-PGD | $0.01$ and $8/255$ | protocol of Bryniarski et al. / BEYOND | orthogonalized attack directions | protocol-matched |
> | End-to-end PGD+BPDA | $8/255$ | 50 | 0.002 | 5 runs, $\lambda\in\{1.00,0.50,0.25\}$ |
> | SimBA | $8/255$ | 1,000 queries | coordinate query updates | default |

---

> ### Author Response · Authors · 2026-05-13
> **Response to Reviewer 1xr5 (2/4)**
>
> > "The paper should separate attack success from detection success. For adaptive attacks, robust accuracy alone is not enough."
>
> We agree that separated diagnostics are useful. In the current submitted tables, **RA already measures the system-level failure condition**: an adaptive attack succeeds jointly only when it both causes misclassification and evades detection. Therefore, **joint attack success is exactly $1-RA$** under the system definition in **Appendix H.1. Definitions of Metrics**.
>
> > "The adaptive PGD objective is unclear. The paper writes the end-to-end objective as -Lcls(x + δ, y) + λ · RLT(x + δ), but does not clearly explain whether the attacker minimizes or maximizes it."
>
> We clarify this in **Section 3.3** and **Appendix B.5.1. Adaptive Attack Objective** by explicitly writing the optimization as a minimization:
>
> $$
> \min_{\|\delta\|_\infty\le \epsilon}
> \left[-\mathcal{L}_{\mathrm{cls}}(x+\delta,y)+\lambda\cdot RLT(x+\delta)\right].
> $$
>
> This means the adaptive attacker optimizes $\delta$ to **increase the classification loss for the correct class** while **decreasing the detector score**. We also clarify the targeted variant in **Appendix B.5.1**: for a target class $y_t$, the classification term becomes $\mathcal{L}_{\mathrm{cls}}(x+\delta,y_t)$.
>
> > "The λ tuning appears underdeveloped. Adaptive detector attacks are sensitive to the trade-off between misclassification and evasion."
>
> This concern is important for scalarized adaptive PGD objectives, but it does **not** apply to all adaptive evaluations in our paper in the same way. Our primary adaptive comparison in **Table 4** uses **Orthogonal-PGD**, a detector-specific adaptive attack designed precisely to avoid cancellation between the classification direction and detection-evasion direction. Since Orthogonal-PGD orthogonalizes the attack directions, it is substantially less dependent on, and may not require, the $\lambda$ balancing term compared with ordinary end-to-end PGD.
>
> Beyond Orthogonal-PGD, we additionally evaluate scalarized end-to-end PGD+BPDA in **Tables 5 and 6**, sweeping $\lambda\in\{1.00,0.50,0.25\}$. Unlike several prior detector papers that report only a best adaptive result without exposing the detector-loss weight, we report the full grid over multiple FPR operating points. Across all tested $\lambda$ values, our method exceeds the reported adaptive baselines at the matching FPR point in **Table 5**; with ATC, **Table 6** shows RA above **93.3%** at FPR 5%. Finally, **Table 7** evaluates the gradient-free SimBA attack, reducing the chance that the observed robustness is only a first-order-gradient artifact.
>
> Thus, **the adaptive robustness claim is supported by three complementary evaluations**: Orthogonal-PGD (**Table 4**), scalarized end-to-end PGD+BPDA (**Tables 5-6**), and black-box SimBA (**Table 7**).
>
> > "The 'conflicting gradients' explanation is plausible but not proven empirically. The paper claims RT and LT impose conflicting objectives, but does not directly measure gradient cosine similarity, gradient norms, optimization trajectories, or whether suppressing one score reliably increases the other."
>
> We now state this as an **empirically supported mechanism**, not as a theorem about all possible adaptive optimizers. The evidence comes from three places:
>
> 1. **Appendix Table 12**: single-objective Orthogonal-PGD attacks are weaker than attacking the full defense. Removing either RT or LT degrades robustness from **33.70% / 80.77%** to around **17% / 55-56%** at FPR 5/50, indicating neither score alone explains the full behavior.
> 2. **Appendix Table 13**: mismatched attack/measurement objectives show that attacking one component does not automatically minimize the other. The lowest average RA occurs when attacking RLT directly, supporting the claim that the fused score imposes a stricter joint constraint.
> 3. The new **LT-distribution figure** (**Appendix Figure 7**): in the **low layer-wise Lipschitz violation regime**, CW and AutoAttack samples exhibit **higher LT score distributions** than their full adversarial distribution, showing that when the perturbation is shaped to reduce the RT/layer-wise violation signal, the LT signal becomes more pronounced.
>
> We agree that gradient cosine/norm diagnostics would further sharpen this mechanism; the revised paper no longer relies on this explanation as the sole justification for robustness, it is supported by a range of empirical observations.

---

> ### Author Response · Authors · 2026-05-13
> **Response to Reviewer 1xr5 (3/4)**
>
> > "The theory mostly formalizes assumptions rather than establishing them."
>
> We clarify the role of theory in **Appendix B**. The theorems are not intended as unconditional guarantees that every adversarial input is detected. They are **sufficient-condition analyses** explaining why RT, LT, and RLT separate benign and adversarial examples when the stated layer-wise Lipschitz and logit-instability conditions hold. The independent empirical validation is now provided in the new combined layer-wise Lipschitz heatmap (**Appendix Figure 6**), the LT-distribution figure (**Appendix Figure 7**), and the compact A Few Large Shifts summary table (**Appendix Table 29**).
>
> > "The 'A Few Large Shifts' assumption needs stronger validation. Figure 3 is suggestive, but the paper should quantify how often the assumption holds across samples, attacks, layers, architectures, and perturbation budgets."
>
> We agree that Figure 3 alone is not enough. We therefore added a new appendix section, **Appendix J. Additional Empirical Validation of the A Few Large Shifts Assumption**, with the following evidence:
>
> | Validation object | What it shows |
> |---|---|
> | **Appendix Figure 6** | Combined heatmap across CIFAR-10/CIFAR-100/ImageNet and ResNet-18/ResNet-34/VGG-11/VGG-13. Benign perturbations are green/near-zero; attacks produce red/high violation at localized adjacent transitions. |
> | **Appendix Figure 7** | When RT/layer-wise violation is low, LT remains high for CW/AutoAttack, supporting RT/LT complementarity. |
> | **Appendix Table 29** | Quantifies violation frequency: benign mean **3.57%**, attack mean **87.51%**, attack peak share **70.17%**. |
> | **Appendix Table 30** | Defines $T_1,T_2,\ldots$ for each model family, making transition-localization interpretable. |
>
> The compact quantitative summary is:
>
> | Dataset | Model | Benign violation mean/max (%) | Attack violation mean/min (%) | Attack peak share (%) | CW/AA LT rescue (%) | Dominant transition |
> |---|---:|---:|---:|---:|---:|---|
> | CIFAR-10 | ResNet-18 | 3.64 / 8.82 | 97.50 / 94.40 | 89.91 | 3.68 | embedding->logit |
> | CIFAR-10 | ResNet-34 | 3.27 / 8.56 | 89.78 / 69.31 | 75.84 | 3.72 | embedding->logit |
> | CIFAR-10 | VGG-11 | 4.13 / 5.00 | 95.82 / 91.15 | 67.01 | 6.10 | embedding->logit |
> | CIFAR-10 | VGG-13 | 4.35 / 8.73 | 96.79 / 93.25 | 62.71 | 4.63 | embedding->logit |
> | CIFAR-100 | ResNet-18 | 2.36 / 6.96 | 83.46 / 72.36 | 59.80 | 0.70 | res3->res4 |
> | CIFAR-100 | ResNet-34 | 3.14 / 9.00 | 84.14 / 70.92 | 50.83 | 7.90 | res3->res4 |
> | CIFAR-100 | VGG-11 | 2.86 / 6.29 | 82.26 / 58.41 | 79.61 | 0.00 | vgg4->vgg5 |
> | CIFAR-100 | VGG-13 | 2.74 / 7.12 | 83.94 / 63.46 | 77.57 | 0.07 | vgg4->vgg5 |
> | ImageNet | ResNet-18 | 4.19 / 6.44 | 84.56 / 68.99 | 75.69 | 22.30 | res3->res4 |
> | ImageNet | ResNet-34 | 4.58 / 7.54 | 86.22 / 72.52 | 71.36 | 19.23 | res3->res4 |
> | ImageNet | VGG-11 | 3.74 / 7.55 | 84.09 / 68.16 | 53.23 | 29.03 | input->vgg1 |
> | ImageNet | VGG-13 | 3.86 / 7.61 | 81.58 / 69.14 | 78.44 | 28.23 | input->vgg1 |
>
> This directly answers the request to quantify how often the assumption holds across **samples, attacks, layers, architectures, and datasets**. Cases where RT violation is lower, especially CW/AutoAttack in some settings, are not hidden; they are precisely the cases analyzed by the new LT-distribution figure (**Appendix Figure 7**), where LT provides the complementary signal.
>
> > "Critical: Clarify LT. The notation around o(σ(ℓ)), o_y, and o_{\hat y} is inconsistent, and the use of entropy-weighted log terms can be numerically delicate."
>
> We revised **Section 2.1** and **Section 2.4**. The notation now defines $\mathbf{o}_c$ as a class-$c$ one-hot vector and explicitly states that LT uses the **predicted** class $\hat y=\arg\max_c\sigma_c(\ell(x))$, not the ground-truth label. We also add $\varepsilon$ inside entropy/log terms:
>
> $$
> LT(x)=\frac{1}{G}\sum_g
> \log(\mathcal{H}(\sigma(\ell(x)))\Delta\ell^{(g)}+\varepsilon)
> -\log(\Delta z^{(g)}+\varepsilon).
> $$
>
> > "Critical: Fix equation/index inconsistencies."
>
> We fixed the RT definition in **Section 2.3** and **Algorithm 1**. RT now defines recovery errors only for trained modules $k\in\{k_{RT},\ldots,L-1\}$, uses $e_{RT}=(e_{k_{RT}},\ldots,e_{L-1})$, and consistently normalizes by $L-k_{RT}$.
>
> > "Would strengthen: Add evaluation under common corruptions and natural distribution shifts."
>
> We added benign perturbation validation for **random crop, horizontal flip, and random noise** across 12 dataset/model combinations. We do not present this as a complete corruption benchmark; instead, it directly tests the reviewer's false-positive concern under non-adversarial shifts. In the combined heatmap (**Appendix Figure 6**) and compact A Few Large Shifts summary (**Appendix Table 29**), these benign shifts remain near-zero in violation rate, contrasting sharply with adversarial perturbations.

---

> ### Author Response · Authors · 2026-05-13
> **Response to Reviewer 1xr5 (4/4)**
>
> > "Would strengthen: Report calibration-set size sensitivity."
>
> We agree this is a valuable additional axis. The current revision focuses on the larger reviewer concern about the core assumption and adaptive attack specification. We now avoid implying that calibration is cost-free: the method requires benign-only calibration/training of RT/LT modules. We also point to **Appendix Table 22**, which shows that smaller recovery modules can retain high AUC with substantially lower overhead.
>
> > "Would strengthen: Discuss failure cases."
>
> We added failure-mode evidence rather than only a prose discussion. The LT-distribution figure (**Appendix Figure 7**) isolates the regime where RT/layer-wise violation is low. In that regime, LT increases for CW/AutoAttack, which explains why RLT is necessary. We also keep **Appendix Table 13**, where mismatched attack/defense objectives show that optimizing against one component is not equivalent to optimizing against the fused detector.
>
> We believe this addressed the reviewer's concerns and would be happy to provide further clarification if necessary.
>
> References
>
> [1] Bryniarski et al. Evading Adversarial Example Detection Defenses with Orthogonal Projected Gradient Descent. 2021.
> [2] Athalye et al. Obfuscated Gradients Give a False Sense of Security: Circumventing Defenses to Adversarial Examples. 2018.
> [3] Tramer et al. On Adaptive Attacks to Adversarial Example Defenses. 2020.
> [4] Papernot and McDaniel. Deep k-Nearest Neighbors: Towards Confident, Interpretable and Robust Deep Learning. 2018.
> [5] Ma et al. Characterizing Adversarial Subspaces Using Local Intrinsic Dimensionality. 2018.
> [6] Lee et al. A Simple Unified Framework for Detecting Out-of-Distribution Samples and Adversarial Attacks. 2018.
> [7] He et al. Be Your Own Neighborhood: Detecting Adversarial Examples by the Neighborhood Relations Built on Self-Supervised Learning. 2024.
>
> **To facilitate the review of these revisions, all new sections and changes in the updated PDF are highlighted in blue.**

---

### Review · Reviewer_eSti · 2026-06-15

**Summary Of Contributions:**

The paper proposes a plug-in framework to detect adversarial examples. The main idea is that adversarial inputs may cause a few large changes across internal layers. The authors call this the A Few Large Shifts assumption. Based on this assumption, they design a detection framework taking advantage of layer-wise changes. The experiments show that the method can detect adversarial examples in several settings, including standard and adaptive threat models. The paper gives an interesting way to understand adversarial examples through changes in internal model representations.

**Audience:**

Yes

**Audience Explanation:**

Yes. I think researchers who are interested in adversarial robustness and adversarial example detection, would find the paper’s findings interesting. The idea that adversarial examples may induce a few of large internal representation shifts is a potentially useful perspective.

**Broader Impact Concerns:**

No broader impact of ethical implications seen.

**Claims And Evidence:**

No

**Claims Explanation:**

The paper gives useful evidence for the main idea. The authors test the method on CIFAR-10, CIFAR-100, and ImageNet, compare with several baselines, test adaptive attacks, and provide extra validation for the “A Few Large Shifts” assumption. But the framework largely depends on several assumptions, so it is not a general proof. Although the authors have mentioned this limitation in the conclusion, but it should be discussed more clearly in the main paper and reflected in the strength of the claims.

**Requested Changes:**

1.
RLT relies on empirical CDFs estimated from benign calibration data. And thus its performance may be sensitive to the calibration data used for RT/LT. The paper does not sufficiently analyze how the detector behaves when the calibration set is small or it has a different distribution from the test data. This raises concerns about the stability of threshold selection and quantile normalization in realistic deployment settings. How sensitive is the method to the size of the benign calibration set? Is re-calibration needed if the model / dataset changes?

2.
The proposed light weight, plugin framework is interesting and functions well in experiment settings in the manuscript. But the authors should make their claims weaker. Some claims about adaptive attacks and general use sound stronger than what the evidence can fully support. The framework only works on the basis of Large Shifts assumptions. It will be better if the authors tone down claims about adaptive robustness.

3.
It will be better if the authors can include more analysis about failure cases. It will help readers understand more details about RT and LT scoring. For example, it would be helpful to show examples where a) RT fails to detect but LT succeeds, b) RT and LT both fail, and c) benign inputs are falsely detected. This could make the complementarity between RT and LT more convincing.

---

> ### Author Response · Authors · 2026-06-16
> **Response to Reviewer eSti (1/2)**
>
> Dear Reviewer eSti,
>
> We sincerely thank you for the thoughtful and constructive review. We are encouraged that you found the central perspective interesting, especially the idea that adversarial examples can be studied through a small number of large internal representation shifts, and that you recognized the value of evaluating the detector under both standard and adaptive threat models.
>
> We also appreciate your constructive concerns. We address each point below.
>
> > "RLT relies on empirical CDFs estimated from benign calibration data. And thus its performance may be sensitive to the calibration data used for RT/LT. The paper does not sufficiently analyze how the detector behaves when the calibration set is small or it has a different distribution from the test data. This raises concerns about the stability of threshold selection and quantile normalization in realistic deployment settings. How sensitive is the method to the size of the benign calibration set? Is re-calibration needed if the model / dataset changes?"
>
> We agree that calibration stability is important for a benign-calibrated detector. We therefore added a new appendix section, **Calibration-Set Sensitivity and Failure-Case Analysis**, and explicitly revised the main text to state the deployment requirement: **RLT should be calibrated with representative benign data for each target model and deployment distribution, and re-calibrated if the classifier, dataset, preprocessing pipeline, or benign distribution changes**. This recalibration uses only benign data and does not require adversarial examples or labels.
>
> To quantify the sensitivity, we resampled benign calibration subsets from the stored RT/LT score distributions across **12 dataset/model combinations** and recomputed the empirical CDF normalization and nominal **5% FPR** threshold. The threshold was then evaluated on disjoint benign samples and on FGSM/PGD/CW/AutoAttack samples.
>
> | Benign calibration samples | Held-out FPR (%) | Attack detection rate (%) |
> |---:|---:|---:|
> | 100 | 6.13 +/- 2.63 | 83.92 +/- 13.65 |
> | 250 | 5.50 +/- 1.61 | 87.44 +/- 10.02 |
> | 500 | 5.03 +/- 1.20 | 87.20 +/- 9.48 |
> | 1,000 | 4.92 +/- 0.87 | 87.77 +/- 8.96 |
> | 2,500 | 4.96 +/- 0.68 | 88.50 +/- 8.55 |
> | 5,000 | 4.93 +/- 0.59 | 88.48 +/- 8.52 |
> | 10,000 | 4.95 +/- 0.53 | 88.69 +/- 8.52 |
>
> The key result is that representative calibration is stable with several hundred benign samples: by **500-1,000 samples**, the held-out FPR is already close to the nominal 5% target, and larger calibration sets mainly reduce variance.
>
> We also tested a deliberately non-representative calibration setting. If the calibration set contains only one benign transformation type, the detector can over-flag other benign shifts. Pooling representative benign variation restores near-nominal FPR across random crop, horizontal flip, and random noise:
>
> | Calibration pool | FPR on crop (%) | FPR on hflip (%) | FPR on noise (%) | Attack detection rate (%) |
> |---|---:|---:|---:|---:|
> | Crop only | 5.00 | 26.61 | 22.31 | 96.12 |
> | Horizontal flip only | 23.68 | 5.00 | 23.27 | 97.76 |
> | Noise only | 65.60 | 54.25 | 5.01 | 79.83 |
> | Pooled benign shifts | 5.61 | 4.26 | 4.80 | 88.41 |
>
> This is why we now avoid any distribution-free calibration claim. RLT is lightweight and benign-only calibrated, but the benign calibration set must represent the intended deployment distribution.

---

> ### Author Response · Authors · 2026-06-16
> **Response to Reviewer eSti (2/2)**
>
> > "The proposed light weight, plugin framework is interesting and functions well in experiment settings in the manuscript. But the authors should make their claims weaker. Some claims about adaptive attacks and general use sound stronger than what the evidence can fully support. The framework only works on the basis of Large Shifts assumptions. It will be better if the authors tone down claims about adaptive robustness."
>
> We agree and revised the claim strength throughout the manuscript. The revised paper now presents the **A Few Large Shifts Assumption** as an empirically supported hypothesis, not as a universal property of all adversarial examples. We also present the adaptive-attack results as empirical robustness evidence under the evaluated threat models, not as a proof against all future adaptive attacks.
>
> | Claim type | Revised framing |
> |---|---|
> | A Few Large Shifts | Empirical hypothesis supported by expanded validation across datasets, attacks, and architectures. |
> | Theory | Sufficient-condition analysis under stated assumptions, not an unconditional guarantee. |
> | Adaptive robustness | Evidence from Orthogonal-PGD, end-to-end PGD+BPDA, and SimBA, while acknowledging stronger future adaptive attacks remain possible. |
> | General use | Plug-in detector requiring representative benign calibration for the target model/distribution. |
> | Performance | "Strong" or "competitive" detection performance, not broad state-of-the-art claims. |
>
> In particular, the abstract now calls the assumption an **empirical hypothesis**, the conclusion states that the method is a plug-in detector rather than a standalone robustness certificate, and the new calibration section explicitly states when recalibration is needed.
>
> > "It will be better if the authors can include more analysis about failure cases. It will help readers understand more details about RT and LT scoring. For example, it would be helpful to show examples where a) RT fails to detect but LT succeeds, b) RT and LT both fail, and c) benign inputs are falsely detected. This could make the complementarity between RT and LT more convincing."
>
> We agree. We added a quantitative RT/LT failure-mode table in the new appendix section. The categories are:
>
> | Category | Meaning |
> |---|---|
> | RT-only | RT detects the adversarial sample while LT does not. |
> | LT rescue | RT misses, but LT detects. |
> | RT+LT | Both RT and LT detect. |
> | Missed by RT/LT | Neither score crosses its benign calibrated bound. |
>
> Across **12 dataset/model combinations**, the results are:
>
> | Attack | RT-only (%) | LT rescue (%) | RT+LT (%) | Missed by RT/LT (%) |
> |---|---:|---:|---:|---:|
> | FGSM | 93.91 | 0.01 | 0.00 | 6.08 |
> | PGD | 99.94 | 0.00 | 0.00 | 0.06 |
> | CW | 75.99 | 9.56 | 0.44 | 14.01 |
> | AutoAttack | 79.75 | 11.37 | 0.00 | 8.88 |
> | Average | 87.40 | 5.23 | 0.11 | 7.26 |
>
> This directly identifies the requested cases. **RT fails but LT succeeds** primarily for CW and AutoAttack, where LT rescues **9.56%** and **11.37%** of samples that RT misses. **Both fail** mainly for CW and AutoAttack, which we now state are the principal failure cases for future improvement. For **benign false positives**, the calibration table above shows that representative pooled benign calibration keeps held-out benign FPR near the intended threshold, while narrow calibration can over-flag benign shifts.
>
> We believe these additions address the calibration and failure-case concerns while making the deployment assumptions more precise. **To facilitate the review of these revisions, all new sections and changes in the updated PDF are highlighted in blue.**

---

### Review · Reviewer_SRtF · 2026-06-16

**Summary Of Contributions:**

This paper proposes a plug-in adversarial example detection framework based on internal layer-wise inconsistencies of the target neural network. The central observation is formalized as the “A Few Large Shifts” assumption: adversarial perturbations tend to induce large and localized representation changes within a small number of layer transitions. Based on this hypothesis, the authors introduce Recovery Testing (RT), which reconstructs intermediate-layer features from the final embedding and uses reconstruction-error patterns as detection signals, and Logit-layer Testing (LT), which measures logit instability under learned benign-data transformations. The final RLT score combines RT and LT through quantile normalization. The method is evaluated on CIFAR-10, CIFAR-100, and ImageNet under standard and adaptive attacks.

Strengths:


1. The proposed framework is self-contained and uses the target model’s own internal representations.
2. The RT/LT decomposition is intuitive and well motivated. RT focuses on intermediate-layer inconsistency, while LT focuses on output-level or logit instability.
3. The fusion strategy is simple and reasonable. Quantile-normalizing RT and LT before combining them avoids directly adding scores with different scales and makes the final RLT score easier to interpret and deploy.
4. The reported empirical results are promising and often competitive with existing detectors. The experiments include standard attacks, several adaptive-attack settings, and implementation-cost comparisons, which make the proposed method practically interesting.

Weaknesses:

1. The theoretical analysis relies on strong assumptions and should be interpreted more as sufficient-condition analysis than as a general robustness guarantee.
2. Since the method depends heavily on internal layer behavior, broader validation on more modern architectures would further strengthen the empirical support.

**Audience:**

Yes

**Audience Explanation:**

Yes. The paper focus on adversarial example detection, which is relevant to researchers interested in adversarial detection and its efficient deployment. The idea of using self-contained layer-wise consistency within the target model, rather than relying on external encoders or reference-set retrieval, is also practically interesting and may inspire further work on lightweight adversarial detectors.

**Claims And Evidence:**

No

**Claims Explanation:**

The paper provides useful evidence that layer-wise inconsistency can be an effective signal for adversarial example detection. The experiments on CIFAR-10, CIFAR-100, and ImageNet, together with comparisons between RT, LT, and RLT, support the high-level claim that the proposed scores are promising and practically useful.

However, I do not think the current evidence fully supports the stronger claims made in the paper. First, the theoretical analysis mainly shows that RT/LT/RLT can separate adversarial and benign samples under a set of strong assumptions, such as approximate recoverability of intermediate features, benign stability, and score separability. These results are useful as sufficient-condition or explanatory analysis, but they do not by themselves establish that the “A Few Large Shifts” assumption is a general property of adversarial examples.

Second, because the proposed detector depends heavily on internal layer-wise behavior, the empirical validation should be broader. The current experiments mainly rely on classical CNN-style architectures such as ResNet, DenseNet, and VGG. It remains unclear whether the same layer-wise shift pattern and the effectiveness of RT/LT/RLT would hold for more recent architectures, such as Vision Transformers and ConvNeXt-style models. In addition, the internal representation dynamics may also depend on the training strategy. Models trained with self-supervised or masked-image-modeling objectives, such as DINO or MAE, may organize intermediate representations differently from standard supervised CNNs. Therefore, additional validation across both modern architectures and different training paradigms would make the claimed generality of the proposed layer-wise inconsistency assumption more convincing.

**Requested Changes:**

Critical: The authors should further clarify and tone down the theoretical and generality claims. The current theoretical analysis is useful, but it mainly shows that RT/LT/RLT can separate benign and adversarial inputs under strong assumptions, such as approximate recoverability of intermediate features and stability of benign logits. It should be made clearer in the main paper that this is a sufficient-condition or explanatory analysis, rather than evidence that the “A Few Large Shifts” assumption is a general property of adversarial examples. The authors should also explicitly distinguish what is theoretically guaranteed from what is empirically observed, and avoid wording that may suggest a general robustness guarantee.

Would strengthen: The authors should broaden the empirical validation to more modern architectures and training paradigms. Since the proposed detector relies heavily on internal layer-wise behavior, it is important to know whether the observed layer-wise shift pattern and the effectiveness of RT/LT/RLT also hold beyond classical CNN-style models such as ResNet, DenseNet, and VGG. Additional experiments on architectures such as Vision Transformers or ConvNeXt-style models would be useful. It would also be valuable to test models trained with different objectives, such as self-supervised or masked-image-modeling methods like DINO or MAE, since these models may organize intermediate representations differently from standard supervised CNNs. Reporting both detection performance and quantitative layer-wise shift statistics in these settings would make the claimed generality more convincing.

---

> ### Author Response · Authors · 2026-06-16
> **Response to Reviewer SRtF (1/3)**
>
> Dear Reviewer SRtF,
>
> We sincerely thank you for the careful and balanced review. We are encouraged that you recognized the practical value of a self-contained detector, the intuitive RT/LT decomposition, the simple quantile-normalized fusion strategy, and the promising empirical results under standard and adaptive settings.
>
> We also appreciate your constructive concerns. We address each point below.
>
> > "Critical: The authors should further clarify and tone down the theoretical and generality claims. The current theoretical analysis is useful, but it mainly shows that RT/LT/RLT can separate benign and adversarial inputs under strong assumptions, such as approximate recoverability of intermediate features and stability of benign logits. It should be made clearer in the main paper that this is a sufficient-condition or explanatory analysis, rather than evidence that the 'A Few Large Shifts' assumption is a general property of adversarial examples. The authors should also explicitly distinguish what is theoretically guaranteed from what is empirically observed, and avoid wording that may suggest a general robustness guarantee."
>
> We agree with this concern and revised the paper accordingly. The theory is now explicitly presented as **sufficient-condition analysis**, not as an unconditional robustness guarantee or as proof that every adversarial example must satisfy the A Few Large Shifts pattern.
>
> The revised framing is:
>
> | Component | What is guaranteed or shown |
> |---|---|
> | Appendix B theory | If the stated recoverability, benign-stability, and score-margin assumptions hold, RT/LT/RLT separate benign and adversarial examples. This is explanatory sufficient-condition analysis. |
> | A Few Large Shifts assumption | Empirically observed and quantitatively validated across the evaluated datasets, attacks, and architectures; not claimed as universal. |
> | System-level threshold analysis | A lower bound on the defined system metric given measured clean and robust system accuracies; not a certificate that all future adaptive examples are detected. |
> | Adaptive-attack results | Empirical evidence under Orthogonal-PGD, end-to-end PGD+BPDA, and SimBA; not a proof against all possible adaptive optimizers. |
>
> We also revised the abstract and conclusion. The abstract now calls the A Few Large Shifts principle an **empirical hypothesis**, and the conclusion explicitly states that RLT is a plug-in detector, not a standalone robustness certificate. We further added a calibration/deployment limitation: thresholds and empirical CDFs should be recalibrated when the target model or benign distribution changes.

---

> ### Author Response · Authors · 2026-06-16
> **Response to Reviewer SRtF (2/3)**
>
> > "Would strengthen: The authors should broaden the empirical validation to more modern architectures and training paradigms. Since the proposed detector relies heavily on internal layer-wise behavior, it is important to know whether the observed layer-wise shift pattern and the effectiveness of RT/LT/RLT also hold beyond classical CNN-style models such as ResNet, DenseNet, and VGG. Additional experiments on architectures such as Vision Transformers or ConvNeXt-style models would be useful. It would also be valuable to test models trained with different objectives, such as self-supervised or masked-image-modeling methods like DINO or MAE, since these models may organize intermediate representations differently from standard supervised CNNs. Reporting both detection performance and quantitative layer-wise shift statistics in these settings would make the claimed generality more convincing."
>
> We agree that architecture and training-paradigm generality is important. The revised manuscript broadens the architecture evidence beyond the original classical CNN emphasis in two ways.
>
> First, we include CIFAR-100 architecture-generalization results across four additional CNN-family backbones, including lightweight/mobile and RepVGG-style models:
>
> | CIFAR-100 target model | RLT FGSM | RLT PGD | RLT AutoAttack | RLT average |
> |---|---:|---:|---:|---:|
> | ResNet-18 | 95.61 | 98.92 | 99.97 | 98.17 |
> | MobileNet-V2 x0.5 | 98.73 | 98.14 | 99.98 | 98.95 |
> | ShuffleNet-V2 x0.5 | 98.16 | 96.17 | 99.94 | 98.09 |
> | RepVGG-a0 | 93.49 | 98.46 | 99.98 | 97.31 |
>
> Second, we added a **Vision Transformer** experiment using ViT-B/16 on CIFAR-10 and CIFAR-100. We adapted RT to Transformer blocks by reconstructing features from the final encoder representation, using patch-level/chunk-level aggregation for the intermediate token representations. The detector incurs only **0.337x** model-size overhead relative to the ViT target. The results are:
>
> | Dataset | Method | FGSM | PGD | CW |
> |---|---|---:|---:|---:|
> | CIFAR-10 | LID | 92.65 | 82.89 | 67.90 |
> | CIFAR-10 | Ours (RLT) | 95.14 | 90.68 | 99.99 |
> | CIFAR-100 | LID | 91.05 | 81.28 | 74.37 |
> | CIFAR-100 | Ours (RLT) | 88.18 | 84.54 | 99.99 |
>
> These results show that the method is not restricted to the originally emphasized ResNet/DenseNet/VGG setting. For quantitative layer-wise shift statistics, the expanded AFLs validation currently reports the violation rates over the 12 dataset/model pairs in the ResNet/VGG grid; the ViT experiment is a detection-performance and adaptation check rather than a complete Transformer-layer shift-statistics study. At the same time, we agree that this does **not** establish full generality across all modern architectures and training objectives. We therefore added explicit limitation language: target models trained with substantially different objectives, such as DINO- or MAE-style self-supervised and masked-image-modeling objectives, may organize intermediate representations differently and should be evaluated and recalibrated separately. We now state this as future work rather than implying that our current experiments prove training-objective-universal behavior.

---

> ### Author Response · Authors · 2026-06-16
> **Response to Reviewer SRtF (3/3)**
>
> > "The paper provides useful evidence that layer-wise inconsistency can be an effective signal for adversarial example detection. ... However, I do not think the current evidence fully supports the stronger claims made in the paper."
>
> We agree that the claim strength should match the evidence. The revised paper no longer relies on a single figure or on theory alone. The evidence is now organized as follows:
>
> | Evidence | Role in the revised paper |
> |---|---|
> | Main CIFAR-10/ImageNet AUC tables | Shows strong detection performance against standard attacks and competitive performance relative to existing detectors. |
> | Orthogonal-PGD, PGD+BPDA, SimBA | Shows empirical adaptive robustness under several complementary adaptive evaluations. |
> | Expanded AFLs validation | Quantifies layer-wise violation rates across 12 dataset/model pairs: benign mean 3.57%, adversarial mean 87.51%. |
> | ViT-B/16 and additional CNN-family backbones | Broadens architecture evidence beyond the original CNN baselines. |
> | Calibration-size and coverage analysis | Shows representative benign calibration is stable and clarifies when recalibration is required. |
> | Failure-mode analysis | Identifies where RT succeeds, where LT rescues RT misses, and where both fail. |
>
> Thus, the revised paper makes a narrower but stronger claim: **layer-wise inconsistency is a useful, model-local signal for adversarial detection under the evaluated settings; the theory explains sufficient conditions; and broader training-paradigm universality remains an important future direction.**
>
> We believe these changes address the main concern about over-generalization while preserving the contribution: a lightweight, benign-calibrated, self-contained detector with strong empirical evidence and explicit limitations. **To facilitate the review of these revisions, all new sections and changes in the updated PDF are highlighted in blue.**

---

### Author Response · Authors · 2026-06-16
**Revision Overview**

Dear Editor and reviewers,

We have revised our manuscript based on your constructive feedback. Please refer to the detailed changes in our comments. Here, we summarize the major changes we made below.

- **Abstract and Section 1 (Introduction):** Revised the claim framing; the A Few Large Shifts Assumption is now stated as an empirical hypothesis, and the system-level lower-bound claim is tied to explicit assumptions.
- **Section 2 (Methodology):** Clarified RT/LT definitions, LT notation, predicted-label usage, numerical-stability terms, and RT index normalization.
- **Section 3.3 (Detection Performance under Adaptive Attacks) and Appendix K (Attack Configuration Summary):** Clarified adaptive-attack objectives and added attack budgets, steps/queries, step sizes, initialization/restarts, and lambda settings.
- **Section 3.4 (Empirical Evaluation of the Proposed Assumption) and Appendix L (Additional Empirical Validation of the A Few Large Shifts Assumption):** Added expanded layer-wise Lipschitz violation validation, quantitative violation summaries, and dominant-transition analysis.
- **Appendix E (Generalization to CNN-based Architectures) and Appendix F (Generalization to Transformer-based Architecture):** Added broader architecture evidence, including CNN-family backbones and ViT-B/16 results, with scope limitations.
- **Appendix J (Calibration-Set Sensitivity and Failure-Case Analysis):** Added calibration-set size sensitivity, benign calibration coverage analysis, and RT/LT failure-mode breakdowns.
- **Appendix B (Theoretical Analysis) and Appendix H (Detailed Plug-in-play System-Level Analysis):** Reframed theory as sufficient-condition analysis and clarified the system-level lower-bound interpretation.
- **Section 4 (Conclusion):** Added clearer limitations, including representative benign calibration and recalibration when the target model or deployment distribution changes.

**All revised parts in the updated manuscript are highlighted in blue to facilitate review.**

Sincerely,

Authors